# Distinct transcriptomic profile of satellite cells contributes to preservation of neuromuscular junctions in extraocular muscles of ALS mice

Ang Li[1]*, Jianxun Yi[1], Xuejun Li[1], Li Dong[1], Lyle W Ostrow[2], Jianjie Ma[3], Jingsong Zhou[1]*

[1]Department of Kinesiology, College of Nursing and Health Innovation, The University of Texas at Arlington, Arlington, United States; [2]Department of Neurology, Lewis Katz School of Medicine at Temple University, Philadelphia, United States; [3]Department of Surgery, Division of Surgical Sciences, University of Virginia, Charlottesville, United States

*For correspondence:
ang.li3@uta.edu (AL);
jingsong.zhou@uta.edu (JZ)

Competing interest: The authors declare that no competing interests exist.

**Abstract** Amyotrophic lateral sclerosis (ALS) is a fatal neuromuscular disorder characterized by progressive weakness of almost all skeletal muscles, whereas extraocular muscles (EOMs) are comparatively spared. While hindlimb and diaphragm muscles of end-stage SOD1G93A (G93A) mice (a familial ALS mouse model) exhibit severe denervation and depletion of Pax7+satellite cells (SCs), we found that the pool of SCs and the integrity of neuromuscular junctions (NMJs) are maintained in EOMs. In cell sorting profiles, SCs derived from hindlimb and diaphragm muscles of G93A mice exhibit denervation-related activation, whereas SCs from EOMs of G93A mice display spontaneous (non-denervation-related) activation, similar to SCs from wild-type mice. Specifically, cultured EOM SCs contain more abundant transcripts of axon guidance molecules, including *Cxcl12*, along with more sustainable renewability than the diaphragm and hindlimb counterparts under differentiation pressure. In neuromuscular co-culture assays, AAV-delivery of *Cxcl12* to G93A-hindlimb SC-derived myotubes enhances motor neuron axon extension and innervation, recapitulating the innervation capacity of EOM SC-derived myotubes. G93A mice fed with sodium butyrate (NaBu) supplementation exhibited less NMJ loss in hindlimb and diaphragm muscles. Additionally, SCs derived from G93A hindlimb and diaphragm muscles displayed elevated expression of *Cxcl12* and improved renewability following NaBu treatment in vitro. Thus, the NaBu-induced transcriptomic changes resembling the patterns of EOM SCs may contribute to the beneficial effects observed in G93A mice. More broadly, the distinct transcriptomic profile of EOM SCs may offer novel therapeutic targets to slow progressive neuromuscular functional decay in ALS and provide possible 'response biomarkers' in pre-clinical and clinical studies.

## eLife assessment

The manuscript by Jingsong Zhou and colleagues uncovers why the extraocular muscles (EOMs) are preserved while other muscles undergo degenerative changes in amyotrophic lateral sclerosis (ALS). In this work, the authors have used a mouse model of familial ALS that carries a G93A mutation in the Sod1 gene to demonstrate that NaBu treatment partially restores the integrity of NMJ in the limb and diaphragm muscles of G93A mice. The findings of the study offer **important** information that EOMs are spared in ALS because they produce protective factors for the NMJ and, more specifically, factors secreted by EOM-derived satellite cells. While most of the experimental approaches are **convincing**, the use of sodium butyrate (NaBu) in this study needs further investigation, as NaBu

might have a variety of biological effects. Overall, this work may help develop future therapeutic interventions for patients with ALS.

## Introduction

Amyotrophic lateral sclerosis (ALS) is a fatal disorder characterized by progressive motor neuron (MN) loss and skeletal muscle wasting, resulting in respiratory failure and death 3–5 years following diagnosis (*Hardiman et al., 2017*; *Vucic et al., 2014*). While not entirely spared, the extraocular muscles (EOMs) of ALS patients and mutant mouse models exhibit comparatively preserved structure and function and retained neuromuscular junctions (NMJ; *Domellöf, 2019*; *Domellof, 2012*; *Ahmadi et al., 2010*; *Valdez et al., 2012*). The reason for this differential involvement remains unclear, especially considering that eye movements are preferentially involved in other neuromuscular disorders like myasthenia gravis, some muscular dystrophies, and mitochondrial myopathies (*Greaves et al., 2010*; *Soltys et al., 2008*).

While ALS is classically considered predominantly a 'dying-forward' process of motor neurons (MN), accumulated evidence (*Luo et al., 2013*; *Yi et al., 2021*; *Zhou et al., 2019*; *Zhou et al., 2010b*; *Xiao et al., 2015*; *Xiao et al., 2018*; *Dobrowolny et al., 2008*; *Wong and Martin, 2010*; *Loeffler et al., 2016*) support that early muscle dysfunction is an important contributor to ALS pathophysiology. The degree to which 'dying-forward' from the central nerve system or 'dying-back' process (*Dadon-Nachum et al., 2011*; *Fischer et al., 2004*; *Frey et al., 2000*) starting distally at NMJs contributing to ALS progression remains unsettled and may vary in different patient subsets. Regardless of the direction of communication and relative contribution of different cell types to ALS progression (e.g. glia, neurons, myofibers; *Frey et al., 2000*), it is widely accepted that NMJ disassembly and the resulting skeletal muscle denervation is a critical pathogenic event in both patients and animal models (*Campanari et al., 2019*; *Cappello and Francolini, 2017*; *Clark et al., 2016*; *Martineau et al., 2018*).

EOMs differ from limb and trunk muscles in many respects (*Horn and Straka, 2021*). In contrast to other skeletal muscles, the individual motor neurons (MNs) controlling EOMs innervate fewer EOM myofibers, and individual myofibers often have multiple NMJs (*Nijssen et al., 2017*). The multiple innervation phenotype refers to the co-existence of innervated 'en plaque' and 'en grappe' NMJs in a single EOM myofiber. The 'en plaque' type of NMJs are located in the middle portion of myofibers and resemble NMJs seen in other types of muscles. Each EOM myofiber has one 'un plaque' NMJ. 'en grappe' NMJs are a series of smaller, grape-shaped NMJs that are located in the distal portion of some EOM myofibers (*Domellöf, 2019*; *Nijssen et al., 2017*). MNs innervating EOMs express higher level of the $Ca^{2+}$ binding proteins calbindin-D28k and parvalbumin, thus may be more resistant to the $Ca^{2+}$-induced excitotoxicity (*Ince et al., 1993*; *Alexianu et al., 1994*; *Mosier et al., 2000*). Pronounced differences exist between EOM and limb muscles in their transcriptomic profiles (*Fischer et al., 2002*). In addition, a recent proteomic study revealed more than 2500 proteins differentially expressed between EOM and EDL muscles and more than 2000 between EOM and soleus muscles (*Eckhardt et al., 2023*). However, none of muscle-derived neuroprotective factors, including BDNF, GDNF, NT3, and NT4 were expressed (RNA or proteins) significantly higher in EOM myofibers compared to hindlimb myofibers in both mutant G93A and wild-type mice (*Eckhardt et al., 2023*; *Harandi et al., 2014*; *Harandi et al., 2016*). Thus, it is important to look for clues in other cell types that influence the microenvironment around NMJs, such as satellite cells (SCs).

An association between SC function, NMJ integrity, and resultant denervation has been demonstrated in animal models of aging and muscular dystrophy (*Liu et al., 2015b*; *Liu et al., 2017*). Using transgenic mice expressing GFP in the progeny of activated SCs, Liu and colleagues demonstrated that inducible depletion of adult skeletal muscle SCs impaired the regeneration of NMJs (*Liu et al., 2015b*). Furthermore, they found that the loss of SCs exacerbated age-related NMJ degeneration. Preservation of SCs attenuated aging-related NMJ loss and improved muscle performance in mice (*Liu et al., 2017*). SCs from EOMs exhibit superior expansion and renewability (the capability to maintain dividing and undifferentiated stem cells after rounds of regeneration) than their limb counterparts (*Stuelsatz et al., 2015*), inspiring us to compare the phenotypes of SCs derived from EOMs and limb muscles and to conduct transcriptomic profiling of SCs to look for candidate genes that could preserve NMJ integrity in ALS.

In the current study, we found that the integrity of NMJs and the amount of Pax7[+] SCs are maintained in EOMs in end-stage G93A mice, in contrast to the severe denervation and depletion of Pax7[+] SCs in hindlimb and diaphragm muscles. Transcriptomic profiling revealed that the EOM SCs exhibit abundant transcripts of axon guidance molecules, such as *Cxcl12*, along with sustainable renewability compared with SCs from diaphragm and hindlimb muscles. Furthermore, EOM SC-derived myotubes enhance axon extension and innervation in co-culture experiments with spinal motor neurons compared to hindlimb SC-derived myotubes. Overexpressing *Cxcl12* using adeno-associated viral vector (AAV8) in G93A hindlimb SC-derived myotubes recapitulated features of the EOM SC-derived myotubes.

Butyrate, a short-chain fatty acid produced by bacterial fermentation of dietary fibers in the colon, has been shown to extend the survival of ALS G93A mice (*Zhang et al., 2017*). We found that G93A mice fed with 2% sodium butyrate (NaBu) in drinking water decreased NMJ denervation and SC depletion in hindlimb and diaphragm muscles. Adding NaBu to the culture medium improved the renewability of SCs derived from the G93A hindlimb and diaphragm muscles, as well as increasing the expression of *Cxcl12,* resembling EOM SC transcriptomic pattern. Characterizing the distinct EOM SC transcriptomic pattern could provide clues for identifying potential biomarkers in therapeutic trials in both ALS patients and animal models, in addition to identifying therapeutic targets.

## Results

### NMJ integrity and peri-NMJ SC abundance are preserved in EOMs, and NaBu-supplemented feeding reduces NMJ denervation and SC depletion in hindlimb and diaphragm muscles in G93A mice

We investigated the NMJ integrity and peri-NMJ SC abundance by whole-mount imaging of extensor digitorum longus (EDL), soleus, diaphragm, and EOMs derived from WT littermates and G93A mice with or without NaBu-supplemented feeding. Acetylcholine receptors (AChRs) at NMJs were labeled with Alexa Fluor conjugated α-Bungarotoxin (BTX). Axons and axon terminals of motor neurons were labeled with antibodies to neurofilament (NF) and synaptophysin (SYP), respectively (*Figure 1A* and *Figure 1—figure supplement 1A*). In order to quantify the extent of denervation in a categorical manner, NMJs were arbitrarily defined as 'well innervated", 'partially innervated', and 'poorly innervated' (*Figure 1—figure supplement 1A*). Poorly innervated NMJs also generally lacked nearby NF[+] axons. For EOMs, only the 'en plaque' type of NMJs were included in this comparison, because they resemble NMJs in other types of muscles in morphology and spatial distribution. The 'en grappe' type of NMJs were not included.

The relative proportions of well-, partially- and poorly-innervated NMJs in different muscle origins of WT and G93A mice are shown in *Figure 1B* (including both genders). At least 343 (average 758±345 per group) NMJs from muscles (EDL, soleus, diaphragm and EOMs, respectively) dissected from WT or G93A mice with or without NaBu-feeding were examined. The original mouse information (including gender specification) and quantification data are listed in *Figure 1—source data 1*. EDL, soleus and diaphragm muscles, but not EOMs, from end stage G93A mice (4-month-old) exhibited significant decrease in well-innervated NMJs, accompanied by an increase in poorly-innervate NMJs. The increase of 'poorly-innervated NMJ' is slightly less dramatic in diaphragm compared to the hindlimb muscles, meanwhile the increase of 'partially-innervated NMJ' ratio is the most significant in the diaphragm (*Figure 1B*), indicating that denervation process is most severe in hindlimb muscles, slightly milder in diaphragm and barely detectable in EOMs. The data showed in *Figure 1B* have also been replotted to compare the innervation status between male and female mice (*Figure 1—figure supplement 2*). In the term of well- or partially- innervated ratios, there are no significant cross-gender difference observed in our experimental condition, in which the muscle samples were collected at the end stage of the disease, although there is marginally lower 'poorly innervated ratio' in the EDL muscle of G93A female mice compared to G93A male mice.

In the previous study, the NaBu treatment of G93A mice started at the pre-symptomatic age (2~2.5-month-old) significantly delayed ALS progression in G93A mice (*Zhang et al., 2017*). As the treatment for ALS patients is initiated after symptoms appear, we further tested whether NaBu treatment started after the disease onset (at the age of 3 months, 2% NaBu in water for 1 month) was effective in preserving NMJ integrity. Consistent with previous observation, the hindlimb and diaphragm

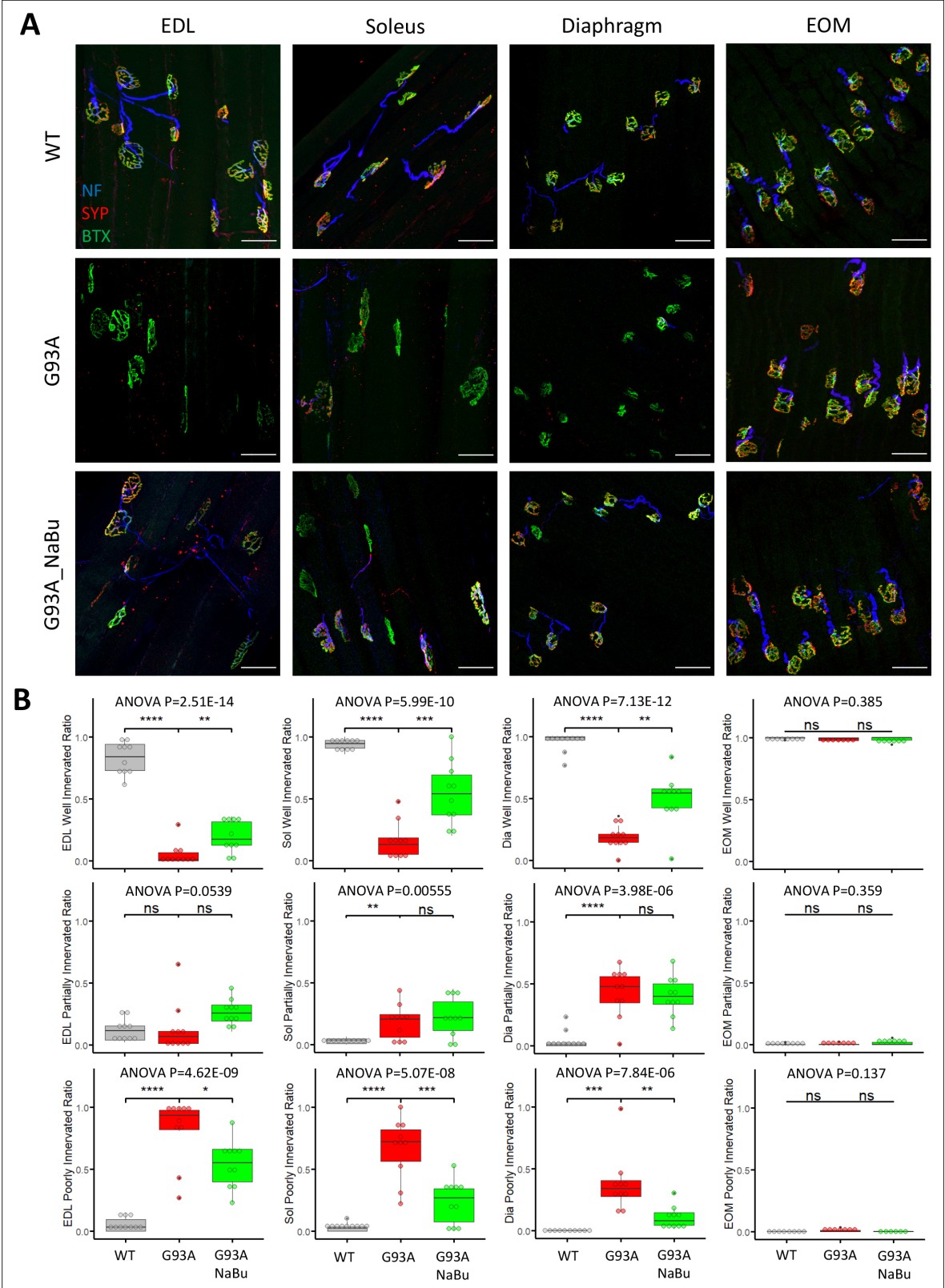

**Figure 1.** Characterizing NMJ integrity in muscles dissected from wild-type controls and end- stage G93A mice with or without NaBu feeding. (**A**) Representative compacted z-stack scan images of whole-mount EDL, soleus diaphragm extraocular muscles stained with antibodies against neurofilament medium chain (NF, labeling axons), synaptophysin (SYP, labeling axon terminals) and Alexa Fluor 488 conjugated α-Bungarotoxin (BTX, labeling AChRs on muscle membrane). scale bars, 50 μm. (**B**) Mean ratios of well innervated NMJs (SYP signals are present in >60% of BTX-positive

*Figure 1 continued on next page*

*Figure 1 continued*

area), partially innervated NMJs (SYP signals are present in 30–60% of BTX positive area) and poorly innervated NMJs (SYP signals are present in <30% of BTX-positive area) and in different types of muscles dissected from WT controls and end-stage G93A mice with or without NaBu feeding (see *Figure 1—figure supplement 1*, *Figure 1—figure supplement 2* and *Figure 1—source data 1* for NMJs measured per muscle type per gender. Briefly, EDL, soleus and diaphragm muscles were from four male and six female mice per group; WT EOM group was from four male and four female mice; G93A EOM group was from three male and four female mice; G93A EOM with NaBu feeding group was from six female mice). Each dot in the box-and-dot plots represents quantification result from a single mouse. * p<0.05; ** p<0.01; **** p<0.0001; ns, not significant (t-test). ANOVA p values are also shown.

The online version of this article includes the following source data and figure supplement(s) for figure 1:

**Source data 1.** Quantification results of the three types of NMJs in muscles of different origins and treatment conditions.

**Source data 2.** qRT-PCR results for *Scn5a* relative expression in whole muscles of different origins and treatment conditions.

**Figure supplement 1.** Representative images of the three types of NMJs and qRT-PCR results of *Scn5a* expression in whole muscles.

**Figure supplement 2.** Cross-gender comparison of NMJ integrity in muscles dissected from wild-type controls and end-stage G93A mice with or without NaBu feeding.

muscles from NaBu-fed mice (1 month, 2% in water, starting to feed at the age of 3-month-old) all exhibited significantly reduced denervation compared to littermates who were not fed NaBu (*Figure 1B*, n=10 per group, gender matched).

qRT-PCR for *Scn5a*, the gene encoding sodium channel Nav1.5, was performed with RNA extracted from whole muscles and used as another marker of denervation (*Sekiguchi et al., 2012*; *Rowan et al., 2012*; *Carreras et al., 2021*; *Figure 1—figure supplement 1B*). The original values of *Scn5a* expression levels are provided in *Figure 1—source data 2*. Consistent with the immunostaining quantification in *Figure 1A*, the elevation of *Scn5a* expression is most notable when comparing G93A hindlimb muscles to wildtype counterparts (>250 folds for EDL, >70 folds for soleus), milder for diaphragm (>15 folds) and not significant for EOMs. Additionally, NaBu feeding significantly reduced the *Scn5a* transcripts in hindlimb muscles (*Figure 1—figure supplement 1B*).

Whole mount imaging of muscles stained with the satellite cell (SC) markers Pax7, BTX, SYP, and DAPI, was used to quantify SCs in the vicinity of NMJs. The Pax7 signal in each z-stack image was filtered using DAPI as a mask to identify Pax7+ cell nuclei. Afterwards the z-stacks were compacted into 2D images by maximal intensity projection (*Figure 2A*). The number of nuclear Pax7+ cells were counted within the circles of 75 μm diameter drawn around the NMJs (*Figure 2B*, see also *Figure 2—source data 1*). Similar to the NMJ denervation status, depletion of peri-NMJ SCs was most significant in hindlimb muscles of end-stage G93A mice and slightly milder in diaphragm, while no significant depletion was observed in EOMs (*Figure 2C*). G93A mice fed with NaBu had less depletion of SCs in hindlimb and diaphragm muscles.

## The proliferation and differentiation properties are conserved in SCs derived from G93A EOMs

We conducted FACS based isolation of SCs from hindlimb (combining tibialis anterior, EDL, quadriceps and gastrocnemius), diaphragm, and EOMs using a modified version of a published protocol (*Liu et al., 2015a*). The three groups were determined because they represent the most severely affected, moderately affected and least affected muscles by ALS progression, respectively. Soleus was not included in the hindlimb SCs pool because its less affected than other hindlimb muscles based on our study and others (*Valdez et al., 2012*; *Atkin et al., 2005*). To separate different cell populations, Vcam1 was used as a marker for quiescent SCs, while CD31 (endothelial cell marker), CD45 (hematopoietic cell marker) and Sca-1 (mesenchymal stem cell marker) positive cells were discarded. Thus, in the FACS profile the target SC population (Vcam1+CD31-CD45-Sca-1-) was in gate P5-4 (*Figure 3A*). Single antibody staining controls are shown in *Figure 3—figure supplement 1*. We also applied methyl green (2 μg/ml) staining (*Prieto et al., 2014*) to evaluate the viability of cells (*Figure 3—figure supplement 2*), only about 1% of Vcam1-positive population were methyl green stained (*Figure 3—figure supplement 2D*, Events P5-2 / (P5–2+P5-4)*100%). Thus, most cells were alive when loaded into the flow cytometer.

SCs isolated from WT hindlimb and diaphragm muscles exhibited relatively homogenous levels of Vcam1, which facilitated separation from other cells and debris (*Figure 3A*, gate P5-4), and also

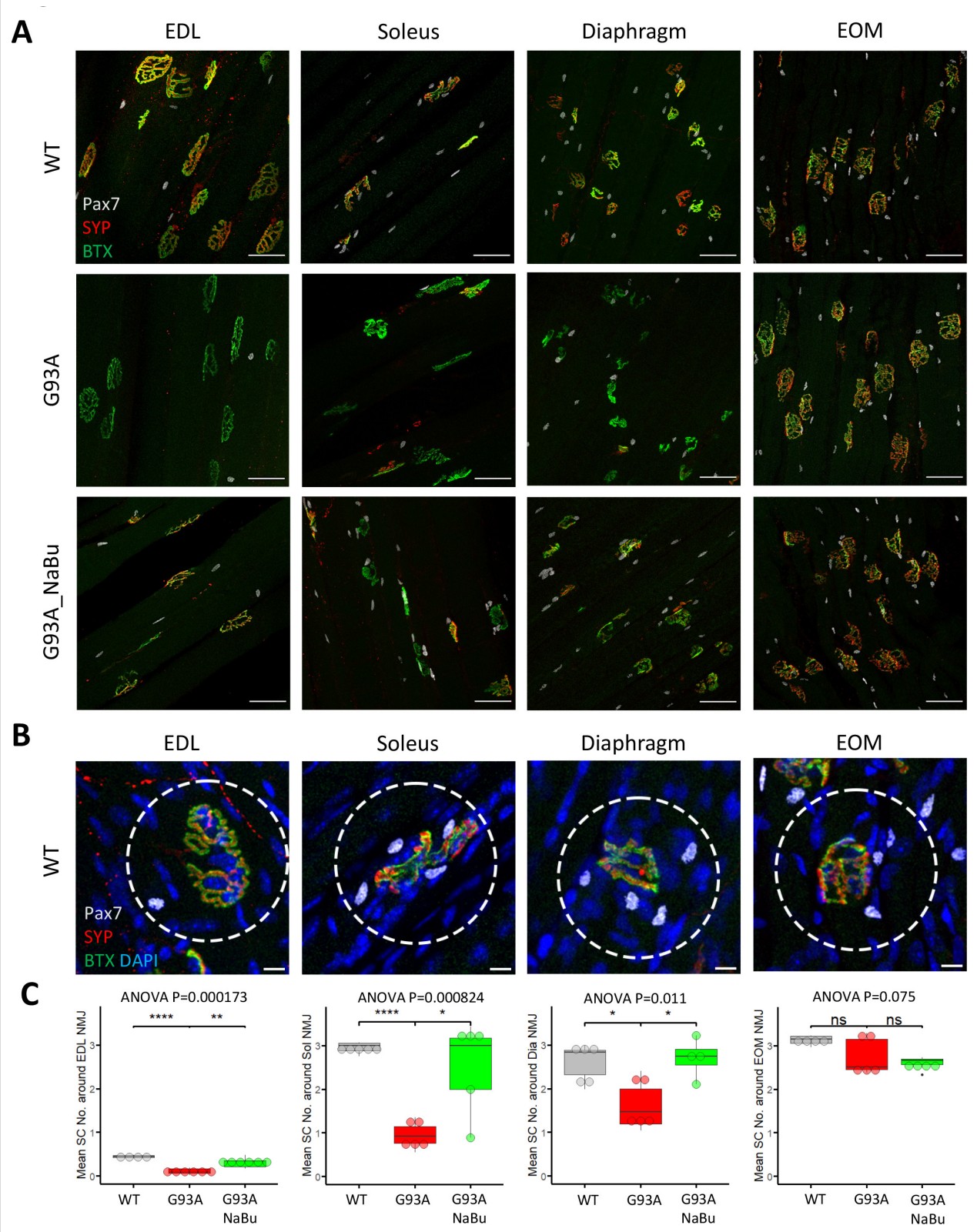

**Figure 2.** Quantification of peri-NMJ SC abundance in muscles dissected from wild-type controls and end-stage G93A mice with or without NaBu feeding. (**A**) Representative compacted z-stack scan images of whole-mount EDL, soleus diaphragm extraocular muscles stained with antibodies against Pax7 (labeling SCs), SYP and BTX-Alexa Fluor 488. Scale bars, 50 µm. (**B**) To measure the abundance of SCs around NMJs, circles of 75 µm diameter were drawn around the NMJs and the number of nuclear Pax7+ cells (co-localized with DAPI) were counted. Scale bars, 10 µm. (**C**) Mean number of

*Figure 2 continued on next page*

*Figure 2 continued*

SCs around NMJs in different types of muscles dissected from WT controls and end-stage G93A mice with or without NaBu feeding (see *Figure 2— source data 1* for NMJs measured per muscle type per gender. Briefly, WT EDL group was from two male and two female mice; G93A EDL group was from three male and three female mice; G93A EDL with NaBu feeding group was from two male and four female mice; WT soleus group was from two male and three female mice; G93A soleus group was from three male and two female mice; G93A soleus with NaBu feeding group was from one male and four female mice; WT diaphragm group was from one male and four female mice; G93A diaphragm group was from one male and four female mice; G93A diaphragm with NaBu feeding group was from four female mice; WT EOM group was from one male and three female mice; G93A EOM group was from five female mice; G93A EOM with NaBu feeding group was from one male and three female mice). Each dot in the box-and-dot plots represents quantification result from a single mouse. * p<0.05; ** p<0.01; **** p<0.0001; ns, not significant (t-test). ANOVA p values are also shown.

The online version of this article includes the following source data for figure 2:

**Source data 1.** Quantification results of averaged peri-NMJ SC numbers in muscles of different origins and treatment conditions.

indicates they are mostly quiescent under resting conditions. In contrast, SCs derived from the G93A hindlimb and diaphragm displayed heterogeneous levels of Vcam1, potentially due to ongoing activation/differentiation triggered by disease progression (*Figure 3A*). Importantly, EOM SCs from both WT and G93A mice were heterogeneous in Vcam1 levels, implying they are spontaneously activated even in the absence of pathological stimulation (*Figure 3A*). While significant decreases of P5-4 population were observed in the G93A hindlimb and diaphragm-derived SCs compared to WT, no significant difference was detected between G93A and WT EOM-derived SCs (*Figure 3B*).

FACS-isolated SCs were next cultured in growth medium for 4 days, fixed, and stained for MyoD to confirm their myogenic lineage. Close to 100% of the cells were MyoD positive (*Figure 4—figure supplement 1A*). We examined the ratio of actively proliferating cells (Ki67$^+$) among the SCs (labeled with Pax7; *Ribeiro et al., 2019*). SCs isolated from both WT and G93A EOMs exhibited the highest Pax/Ki67 double positive ratio, and the lowest Pax7/Ki67 double negative ratio (*Figure 4A*). These observations are consistent with previous reports of superior proliferative potential of EOM SCs compared to limb and diaphragm muscles (*Stuelsatz et al., 2015*). The direct comparison between WT and G93A muscle SCs is presented in *Figure 4—figure supplement 2*.

We also compared myotube formation and cell proliferation of cultured SCs under differentiation conditions. After 4 days in growth medium, we switched to differentiation medium and cultured the cells for 2 more days. The cells were then stained for sarcomere myosin heavy chain (MHC), Pax7 and Ki67. The EOM-derived SCs exhibited a higher number of Pax7-positive cells (Pax7$^+$/Ki67$^+$ and Pax7$^+$/Ki67$^-$) compared to SCs derived from hindlimb and diaphragm muscles – from both WT and G93A mice (*Figure 4B*). In contrast, the fusion indices, which calculate the ratio of nucleus within the MHC-positive myotubes, were the lowest for EOM SCs (*Figure 4B*). This does not necessarily mean the EOM SCs have deficiencies in differentiation after they are committed, but more likely a reflection of their superior preservation of actively proliferating population (*Ancel et al., 2021*). In support of this view, previous TA engraftment experiments showed that more myofibers were generated by EOM SCs than their limb counterparts (*Stuelsatz et al., 2015*).

## Butyrate induces EOM-like preservation of renewability in FACS-isolated SCs under differentiation conditions in vitro

Since NaBu-supplemented feeding reduced SC depletion in end stage G93A mice (*Figure 2*) in vivo, we tested NaBu treatment on cultured SCs. G93A hindlimb SCs were cultured with 0, 0.1, 0.5, 1.0, 2.5, and 5 mM of NaBu added to the growth medium for 1 day. RNAs were collected for qRT-PCR to evaluate the short-term impact of NaBu on cell growth. The relative expression of the proliferation markers (*Hmhg2* and *Ccnd1*) peaked at 0.5 mM. In contrast, the relative expression of proliferation inhibitor P21 bottomed at 0.5 mM (*Figure 5A*), indicating 0.5 mM as the optimal concentration of NaBu application in vitro.

We designed two NaBu treatment plans for G93A diaphragm and hindlimb SCs as follows: one with 0.5 mM NaBu applied in the growth medium for 1 day before switching to differentiation medium (1 day in growth medium), the other with NaBu continuously applied for 3 days (1 day in growth medium +2 days in differentiation medium). As shown in *Figure 5B*, the G93A hindlimb and diaphragm SCs in culture showed significant increase in the ratio of Pax7-positive cells under the 1-day NaBu treatment, associated with a decrease in fusion index. This trend was more pronounced under the 3-day NaBu treatment. These data indicate that NaBu treatment results in EOM-like preservation of

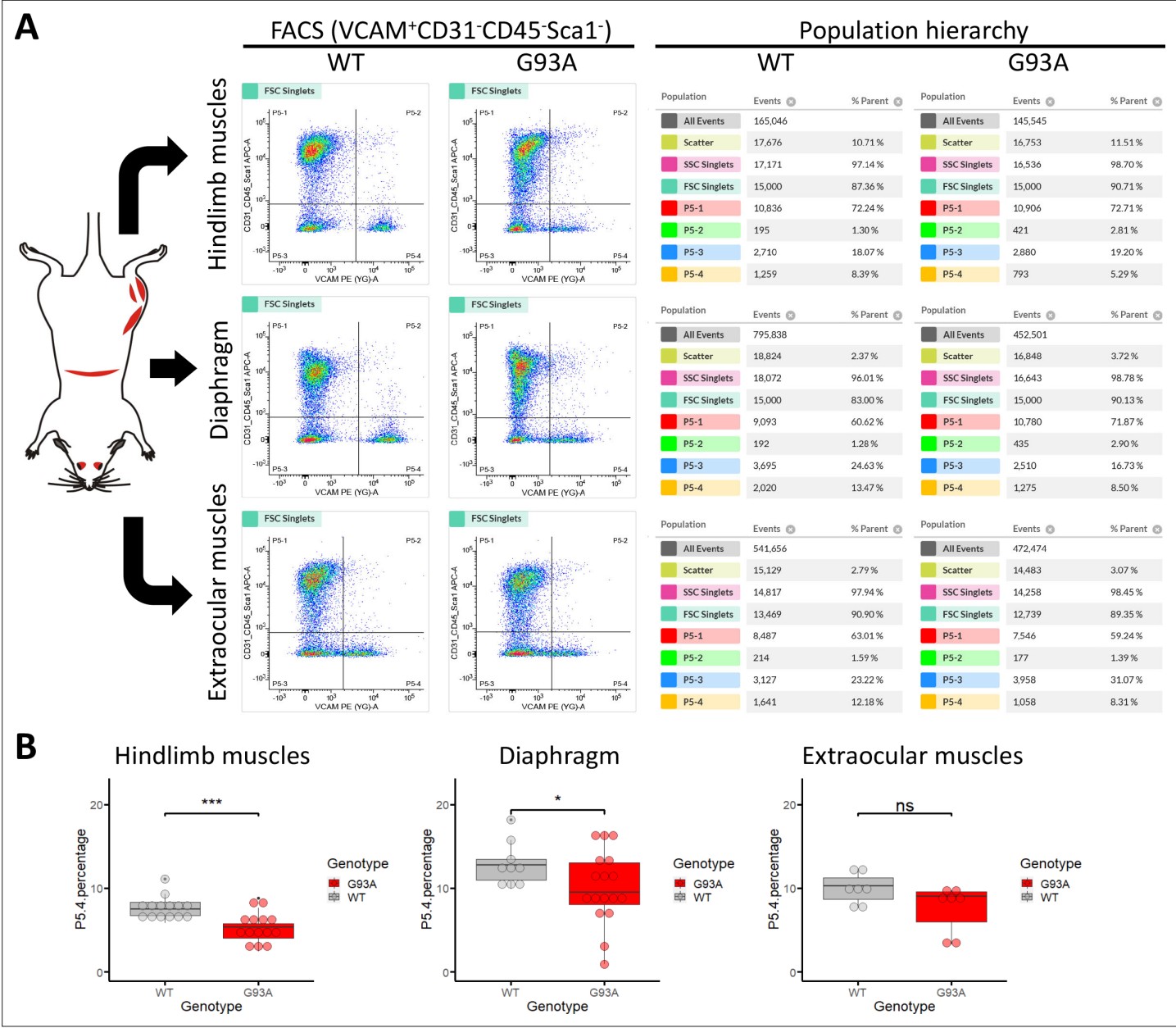

**Figure 3.** Isolation of SCs from wildtype and G93A muscles using fluorescence activated cell sorting. (**A**) Representative FACS profiles during the isolation of Vcam1⁺CD31⁻CD45⁻Sca1⁻ cells (gate P5-4). The corresponding population hierarchies are shown right to the plots. For single antibody staining controls, unstained controls and viability test, see *Figure 3—figure supplement 1* and *Figure 3—figure supplement 2*. (**B**) Comparing the percentage of events in the P5-4 gate between cells isolated from WT and G93A mice. Each dot in the box-and-dot plots represents result from one round of sorting. WT HL SCs were from eight male and six female mice; G93A HL SCs were from nine male and five female mice; WT diaphragm SCs were from six male and three female mice; G93A diaphragm SCs were from 12 male and 5 female mice. WT EOM SCs were from six batches of male and one batch of female mice (each batch contains five to six mice of the same gender). G93A EOM SCs were from five batches of male and two batches of female mice. * p<0.05; *** p<0.001; ns: not significant (t-test).

The online version of this article includes the following source data and figure supplement(s) for figure 3:

**Source data 1.** Percentage of P5-4 events recorded in different rounds of sorting.

**Figure supplement 1.** Representative FACS profiles of single antibody staining controls.

**Figure supplement 2.** Viability assessment of cells for sorting using methyl green.

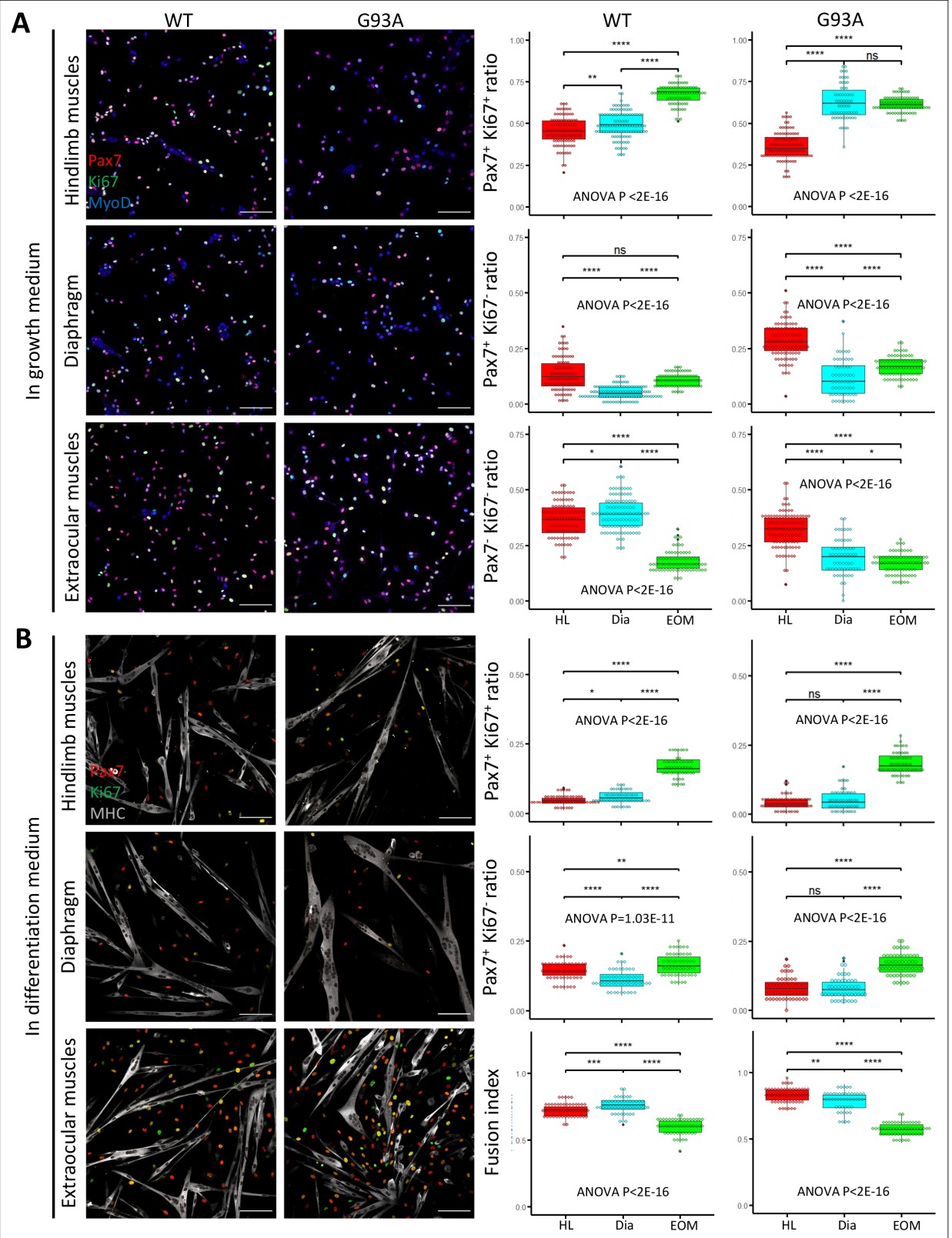

**Figure 4.** Proliferation and differentiation properties of FACS-isolated SCs in culture. (**A**) Representative images of FACS-isolated SCs cultured for 4 days in growth medium stained with antibodies against Pax7, Ki67 (proliferating cell marker), MyoD (myogenic lineage marker). Measurement results for the ratios of Pax7$^+$Ki67$^+$, Pax7$^+$Ki67$^-$, and Pax7$^-$Ki67$^-$ cells are shown in the right two panels. Scale bars, 100 μm. Each dot in the box-and-dot plots represents one image analyzed. For WT HL SCs, 94 images from 3 rounds of sorting; For WT Dia SCs, 107 images from 3 rounds of sorting; For WT EOM

*Figure 4 continued on next page*

*Figure 4 continued*

SCs, 75 images from 3 rounds of sorting; For G93A HL SCs, 96 images from 3 rounds of sorting; For G93A Dia SCs, 62 images from 3 rounds of sorting; For G93A EOM SCs, 79 images from 3 rounds of sorting. For the 3 rounds of sorting, 1 was from male and 2 were from female mice. * p<0.05; ** p<0.01; **** p<0.0001; ns, not significant (t-test). ANOVA p values are also shown. (**B**) Representative images of FACS-isolated SCs cultured for 4 days in growth medium and 2 days in differentiation medium stained with antibodies against Pax7, Ki67, myosin heavy chain (MHC, differentiated myotube marker). Measurement results for the ratios of Pax7$^+$Ki67$^+$ and Pax7$^+$Ki67$^-$ cells, as well as the fusion indices are shown in the right two panels. Each dot in the box-and-dot plots represents one image analyzed. For WT HL SCs, 52 images from 3 rounds of sorting; For WT Dia SCs, 51 images from 3 rounds of sorting; For WT EOM SCs, 51 images from 3 rounds of sorting; For G93A HL SCs, 52 images from 3 rounds of sorting; For G93A Dia SCs, 47 images from 3 rounds of sorting; For G93A EOM SCs, 56 images from 3 rounds of sorting. For the 3 rounds of sorting, 1 was from male and 2 were from female mice. ANOVA P values are also shown.

The online version of this article includes the following figure supplement(s) for figure 4:

**Figure supplement 1.** Enlarged single channel images of SCs cultured in growth and differentiation medium.

**Figure supplement 2.** Comparing proliferation and differentiation properties of cultured SCs from WT and G93A muscles.

renewability of the G93A SCs (*Figure 4B*). Aside from the drop of fusion indices after 3-day treatment with NaBu, there was notable decrease of MHC levels, indicating long-term exposure to NaBu did inhibit the differentiation process (*Figure 5B*), which is in line with the published result observed in myoblasts of different origins (*Iezzi et al., 2002*; *Fiszman et al., 1980*; *Johnston et al., 1992*).

## EOM-derived SCs show distinct transcriptome profile supporting self-renewal and axonal growth when compared to the SCs derived from other muscle origins

We extracted RNAs from SCs of hindlimb, diaphragm and EOMs under the following two culture conditions: first, in growth medium for 4 days (with suffix 'G'); second, in growth medium for 4 days and then in differentiation medium for 2 days (with suffix 'D'). The RNAs were send for bulk RNA-Seq. Additionally, RNA-Seq analyses were conducted for SCs of G93A hindlimb and diaphragm muscles cultured with NaBu for 3 days (with suffix 'NaBu3'; *Figure 6*).

Similarities of the transcriptomic profiles of RNA-Seq samples were examined using the Principle Component Analysis (PCA) and hierarchical clustering (*Figure 6A*). We found that both WT and G93A EOM SCs cultured in differentiation medium were more similar to hindlimb and diaphragm SCs in growth medium than their counterparts in differentiation medium, highlighting the superior renewability of EOM SCs in differentiation medium and is consistent with the immunostaining results showed in *Figure 4B*.

Considering that cultured WT and G93A SCs from the same muscle origins exhibited relatively similar gross transcriptomic profiles in PCA and hierarchical clustering analysis, and that the proliferation and differentiation properties of cultured WT and G93A SCs from the same muscles were only marginally different in immunofluorescence assays (*Figure 4—figure supplement 2*), we speculate that the differential homeostasis seen in the FACS profiles of hindlimb and diaphragm SCs (*Figure 3*) may relate to the pathological milieu in vivo. When G93A SCs are isolated and cultured in vitro, their properties become similar to those of WT counterparts.

Additionally, we listed top 20 differentially expressed genes (DEGs; ranked by Log2FC, both the upregulated and downregulated) by comparing: (1) EOM SCs to their hindlimb and diaphragm counterparts (*Figure 6—source data 1*); (2) G93A SCs to WT SCs of the same muscle origin (*Figure 6—source data 2*); (3) G93A hindlimb and diaphragm SCs with and without 3-day NaBu treatment (*Figure 6—source data 3*).

The top 20 DEGs comparing EOM SCs to their hindlimb and diaphragm counterparts are mainly homeobox transcription factors (*Figure 6—source data 1*), reflecting different anatomic origin of these SCs (*De Robertis et al., 1990*). For example, Alx4 and Pitx1 are expressed significantly higher in EOM SCs, while *Lbx1*, *Tbx1*, *Hoxa*, and *Hoxc* genes are expressed significantly lower in EOM SCs than hindlimb and diaphragm counterparts acquired from either WT or G93A mice. These observations are consistent with previous reports (*Sambasivan et al., 2009*; *Evano et al., 2020*). Additionally, Pax3 is expressed significantly lower in EOM SCs than hindlimb and diaphragm counterparts cultured in growth medium, which reflects the unique Pax3-independent induction of myoD expression reported in EOM SCs (*Sambasivan et al., 2009*). It is also worth mentioning that BMP4 is expressed significantly higher in G93A EOM SCs compared to G93A hindlimb and diaphragm counterparts cultured

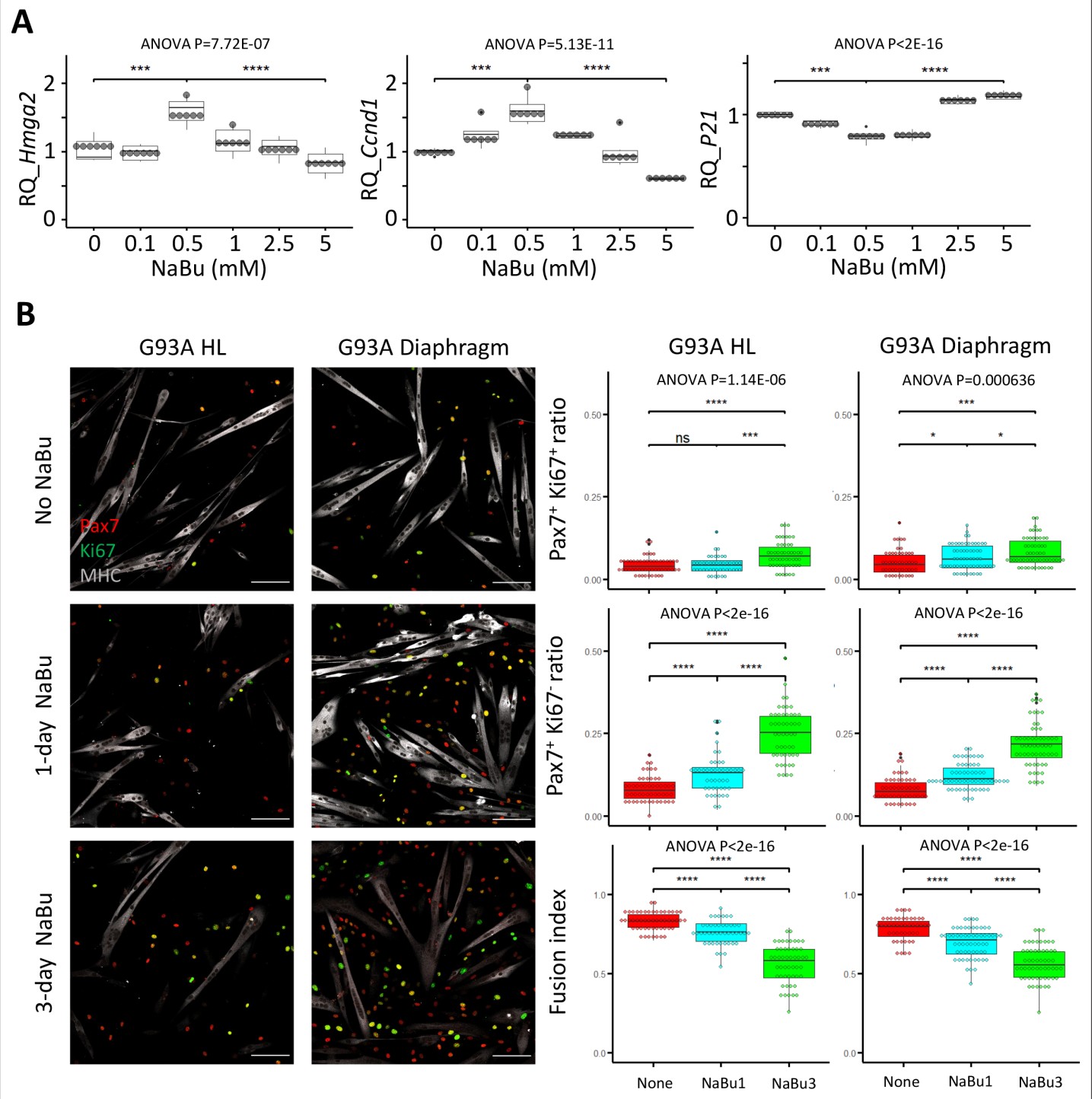

**Figure 5.** Proliferation and differentiation properties of FACS-isolated SCs with different NaBu treatment conditions. (**A**) qRT-PCR-based relative quantification of Hmga2, Ccnd1 and P21 expression in FACS-isolated G93A HL SCs cultured with different doses of NaBu in growth medium for 1 day (n=6, each dot in the box-and-dot plots represents one replicate of culture). HL SCs were from male mice. *** p<0.001; **** p<0.0001 (t-test). ANOVA p values are also shown. (**B**) Left two panels show the representative images of FACS-isolated G93A hindlimb and diaphragm SCs cultured in growth medium for 4 days and differentiation medium for 2 days and experienced different NaBu treatment conditions and stained with Pax7, Ki67, and MHC antibodies. Cells in top row were not treated with NaBu. Cells in the middle row were treated with NaBu for 1 day before the induction of differentiation (Day 4 of culture). Cells in the bottom row were treated with NaBu continuously for 3 days from the day before the induction of differentiation (Day 4–6 of culture). Scale bars, 100 μm. Measurement results for the ratios of Pax7⁺Ki67⁺ and Pax7⁺Ki67⁻ cells, as well as the fusion indices are shown in the right two panels. Each dot in the box-and-dot plots represents one image analyzed. For G93A HL SCs, 52 images from 3 rounds of sorting; 1-day NaBu

*Figure 5 continued on next page*

Figure 5 continued

treatment, 45 images from 3 rounds of sorting; 3-day NaBu treatment, 51 images from 3 rounds of sorting; For G93A Dia SCs, 47 images from 3 rounds of sorting; 1-day NaBu treatment, 60 images from 3 rounds of sorting; 3-day NaBu treatment, 57 images from 3 rounds of sorting. For the 3 rounds of sorting, 2 were from male and 1 was from female mice. * p<0.05; ns, not significant (t-test). ANOVA p values are also shown.

in differentiation medium, potentially contributing to the continued expansion and slowed myogenic differentiation of EOM SCs (*Ono et al., 2011*). This trend is also seen in SCs acquired from WT mice, but not pronounced enough to make it to the top 20 list. However, most of the top 20 differentially expressed genes between G93A and WT SCs (*Figure 6—source data 2*) are with poor functional annotation records and there exist great variations in the lists between muscles of different origins. We also examined the top 20 differentially expressed genes following 3-day NaBu treatment in G93A hindlimb and diaphragm SCs. It did not turn out to be very informative (*Figure 6—source data 3*).

To better understand RNA-Seq data in the perspective of SC homeostasis, which refers to the transition between quiescence (dormant, slow self-renewal), activation (rapid amplification with symmetric and asymmetric cell division) and differentiation states during tissue regeneration (*Ancel et al., 2021*), we compiled the lists of quiescence, activation and differentiation signature genes respectively. For quiescence signature genes, a list of 507 genes highly upregulated in quiescent SCs compared to activated SCs have been generated previously by *Fukada et al., 2007*. We adopted this list together with other 18 genes denoted by other investigators, including *Notch1, Notch2, Hes1, Hey1, Rbpj, Dtx4, Jag1, Sox8, Sdc3, Itga7, Itgb1, Met, Vcam1, Ncam1, Cdh2, Cdh15, Cxcr4, Cav1* (*Ancel et al., 2021*; *Fujimaki et al., 2018*; *Mourikis et al., 2012*; *Bröhl et al., 2012*; *Schmidt et al., 2003*). The activation signature gene list was generated from our RNA-seq data using those expressed significantly higher in SCs (both from WT and G93A) in growth medium (1187 genes in total, |log2FC|≥0.4 and FDR ≤0.001). Accordingly, the differentiation signature gene list was generated from those expressed significantly higher in SCs cultured in differentiation medium (1349 genes in total).

To verify that our signature gene lists recapitulate the SC properties at activation and differentiation state reported in previous studies, we did functional annotation analysis (KEGG pathway, Gene ontology_biological process, Gene ontology_cellular component). As shown in *Figure 6—figure supplement 1*, the top functional annotations enriched in our activation signature genes are all related to cell cycle and DNA replication, while the top functional annotations for our differentiation signature genes are associated with muscle contraction, muscle-specific subcellular structures and metabolic changes.

To explore whether the homeostatic states of EOM SCs are different from that of SCs from other muscle origins, we looked for differentially expressed genes (DEGs) shared-in-common among the following four groups: WT EOM SC-vs-WT diaphragm SC; WT EOM SC-vs-WT HL SC; G93A EOM SC-vs-G93A diaphragm SC; G93A EOM SC-vs-G93A HL SC (*Figure 6B*, left two panels). Within the 1119 shared DEGs identified in SCs cultured in growth medium, slightly more activation signature genes and quiescence signature genes were expressed higher in EOM SCs compared to diaphragm and hindlimb-derived SCs. Meanwhile more differentiation signature genes were expressed lower in EOM SCs (*Figure 6C*, leftmost panel and *Figure 6—source data 4*). These trends became much more prominent in the 3318 shared DEGs identified in SCs cultured in differentiation medium, implying that EOM SCs are notably superior at preserving renewability in a differentiation environment than their diaphragm and hindlimb counterparts (*Figure 6C*, middle panel). Consistently, subgroups of quiescence signature genes belonging to commonly used stem cell markers (*Pax7, Cd34, Cxcr4, Vcam1, Sdc3, Sdc4, Cdh15, Met, Cav1*), Notch signaling pathway (*Notch1, Notch2, Notch3, Jag1, Heyl, Dtx4, Msc*) and pluripotency signaling pathway (*Klf4, Igf1, Bmp2, Bmp4, Fzd4, Fgfr4, Pik3r1, Lifr*) were all expressed higher in EOM SCs in differentiation medium than SCs derived from diaphragm and hindlimb muscles (*Figure 6C*, middle panel and *Figure 6—source data 5*).

In parallel, we examined whether the 3-day NaBu treatment could alter the homeostasis of G93A diaphragm and hindlimb SCs (*Figure 6B*, rightmost panel). There were 3018 DEGs shared between the following two comparing groups: G93A diaphragm SC_D_NaBu3-vs-G93A diaphragm SC_D and G93A HL SC_D_NaBu3-vs-G93A HL SC_D. Similar to the trends observed in EOM SCs in differentiation medium, the majority of activation and quiescence signature genes found in these DEGs were upregulated in the NaBu treated group, while the differentiation signature genes were downregulated (*Figure 6C*, rightmost panel). The elevated quiescence signature genes of the NaBu treated

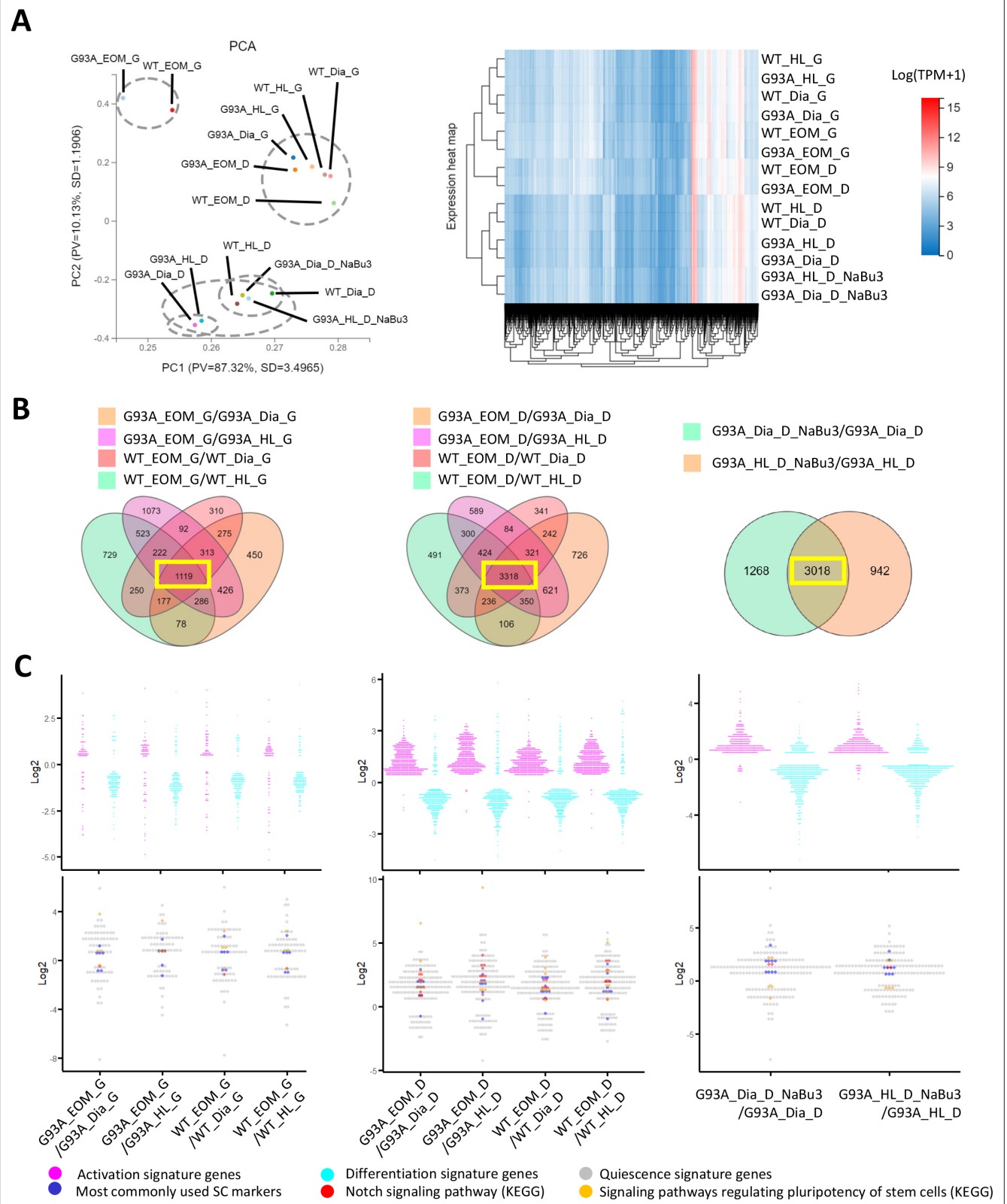

**Figure 6.** Comparing the transcriptomic profile and homeostasis preferences of SCs from different muscle origins cultured in growth and differentiation medium. (**A**) PCA analysis and hierarchical clustering results of different RNA-Seq samples. Samples with suffix 'G' were SCs cultured in growth medium for 4 days. Suffix 'D' denote SCs cultured in growth medium for 4 days and differentiation medium for 2 days. Suffix 'NaBu3' denote SCs treated with 0.5 mM NaBu continuously for 3 days (Day 4–6 of culture). Dashed-line circles in PCA plot highlight samples close to each other in distance. All

*Figure 6 continued on next page*

*Figure 6 continued*

SC samples shown here were from female mice. (**B**) We identify genes differentially expressed in EOM SCs compared to their diaphragm and HL counterparts cultured in growth medium (leftmost panel) and differentiation medium (middle panel) by screening for genes shared by the four DEG lists (gold, magenta, pink, and green in color) shown in the Venn diagrams. All DEG lists were generated based on the standard of |log2FC| ≥ 0.4 and FDR ≤ 0.001. Yellow box highlights the shared genes. To identify NaBu treatment signature genes, we screened for genes shared by the two DEG lists (green and gold in color) shown in the rightmost Venn diagram. (**C**) Dot plots in the upper row show the Log2FC of activation signature genes (magenta) and differentiation signature genes (cyan) identified in the 1119, 3318, and 3018 genes highlighted in yellow boxes in (**B**). Dot plots in the lower row show the Log2 FC of quiescence signature genes identified in the yellow-box highlighted genes in (**B**). Additionally, genes belonging to the following subgroups of quiescent signature genes: most commonly used SC markers, Notch signaling components and pluripotency signaling components are colored in dark blue, red, and gold, respectively.

The online version of this article includes the following source data and figure supplement(s) for figure 6:

**Source data 1.** Top 20 differentially expressed genes comparing EOM SCs to hindlimb and diaphragm counterparts cultured in growth and differentiation medium.

**Source data 2.** Top 20 differentially expressed genes comparing G93A to WT SCs of the same muscle origin cultured in growth and differentiation medium.

**Source data 3.** Top 20 differentially expressed genes comparing G93A hindlimb and diaphragm SCs with or without 3-day NaBu treatment.

**Source data 4.** Three subgroups of quiescent signature genes identified in the DEG lists comparing EOM SCs to diaphragm and HL counterparts cultured in differentiation medium.

**Source data 5.** Three subgroups of quiescent signature genes identified in the DEG lists comparing EOM SCs to diaphragm and HL counterparts cultured in growth medium.

**Source data 6.** Three subgroups of quiescent signature genes identified in the DEG lists comparing G93A diaphragm and HL SCs with and without 3-day NaBu treatment.

**Figure supplement 1.** Compiling the activation signature gene and differentiation signature gene lists.

group include commonly used stem cell markers (*Pax7, Cd34, Vcam1, Sdc4, Cdh2, Met, Cav1, Itga7*), Notch signaling pathway members (*Notch3, Rbpj*) and pluripotency signaling pathway components (*Fzd4, Bmp4*). However, other three pluripotency-related genes (*Klf4, Wnt4* and *Bmp2*) were found downregulated (*Figure 6—source data 6*), indicating that NaBu treatment also has a mixed impact on the stemness of the SCs.

To compile EOM SCs signature gene list, we identified 621 genes expressed higher in EOM SCs than diaphragm and hindlimb SCs cultured in growth medium from the 1119 DEGs (*Figure 6B*, leftmost panel). Next, we pulled out 2424 genes that are expressed higher in EOM SCs than diaphragm and hindlimb SCs cultured in differentiation medium from the 3318 DEGs (*Figure 6B*, middle panel). These two lists have 478 genes shared-in-common. Particularly, many of these genes are related to axon guidance, cell migration and neuron projection (*Figure 7A*). *Figure 7—figure supplement 1A*, *Figure 7—source data 1 and 2* further demonstrate this phenotype when the functional annotation analysis was done separately for the growth or differentiation culture conditions. The functional annotation analysis for the 3018 genes altered by the 3-day NaBu treatment (*Figure 6B*, rightmost panel) also spotted axon guidance, cell migration and neuron projection on the top of the annotation lists (*Figure 7B* and *Figure 7—source data 3*). Importantly, the chemokine *Cxcl12* shows up in all three lists. Cxcl12 is known to promote axonal extension and NMJ regeneration (*Negro et al., 2017*). These data suggest that enhanced attraction of motor neuron axons by peri-NMJ SCs might be a shared mechanism underlying the NMJ preservation in EOMs and NaBu treatment observed in G93A mice.

We selected one representative gene from each of the following categories: activation (*Hmga2*), differentiation (*Actn3*) and quiescence (*Notch3*) signature genes, as well as an axon guidance gene (*Cxcl12*) for qRT-PCR. For each muscle type and culture condition, we collected samples from three to six rounds of sorting (the individual results are listed in *Figure 7—source data 4–6*). We observed significantly higher expression of *Hmga2* in EOM SCs compared to diaphragm and hindlimb SCs cultured in either growth or differentiation medium (*Figure 7C*, leftmost panel), confirming the superior renewability of EOM SCs. The 3-day NaBu treatment of diaphragm and hindlimb SCs derived from G93A mice also upregulated *Hmga2* (*Figure 7D*, leftmost panel). *Actn3* was expressed lower in EOM SCs than diaphragm SCs cultured in either growth or differentiation medium, but not necessarily lower than hindlimb SCs (*Figure 7C*, 2nd panel from the left). Meanwhile NaBu treatment significantly decreased *Actn3* expression (*Figure 7D*, 2nd panel from the left). Consistent with the RNA-Seq data, the quiescence marker *Notch3* was expressed significantly higher in EOM SCs than diaphragm and

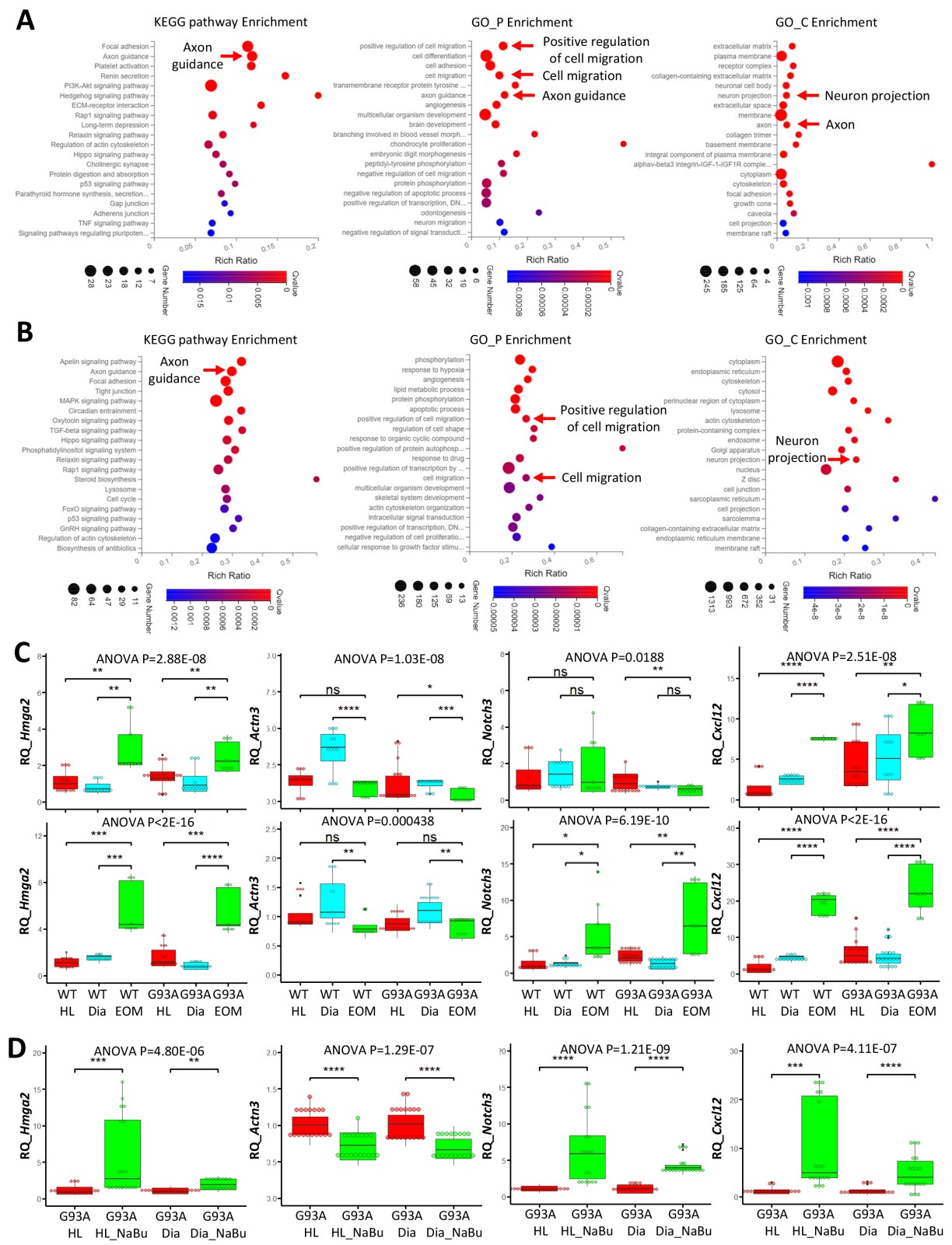

**Figure 7.** Functional annotation analysis of EOM SC signature genes and NaBu treatment signature genes and qRT-PCR results. (**A**) Top KEGG pathway, GO_P and GO_C annotations of the 478 EOM signature genes (expressed higher in EOM SCs compared to diaphragm and HL counterparts cultured in both growth medium and differentiation medium). Arrows highlight annotations related to axon guidance and cell migration. (**B**) Top KEGG pathway, GO_P and GO_C annotations of the 3018 NaBu treatment signature genes shown in **Figure 6B**, rightmost panel. (**C**) qRT-PCR based relative

*Figure 7 continued on next page*

*Figure 7 continued*

quantification (RQ, fold of change against WT HL SCs) of *Hmga2* (activation signature gene), *Actn3* (differentiation signature gene), *Notch3* (quiescence signature gene), and *Cxcl12* (axon guidance molecule) expression in HL, diaphragm and EOM SCs cultured in growth (upper row) and differentiation medium (lower row), respectively. RNA samples were collected from three to six rounds of sorting and sorted cells were seeded into three dishes as replicates. Each dot in the box-and-dot plots represents one replicate of culture. WT HL SCs were from three male and one female mice. WT diaphragm SCs were from two male and two female mice; WT EOM SCs were from three male mice; G93A HL SCs were from four male and two female mice. G93A diaphragm SCs were from one male and three female mice; G93A EOM SCs were from three male mice. * p<0.05; ** p<0.01; *** p<0.001; **** p<0.0001; ns, not significant (t-test). ANOVA p values are also shown. (D) qRT-PCR based relative quantification (fold of change against G93A HL SCs without NaBu treatment) of *Hmga2*, *Actn3*, *Notch3,* and *Cxcl12* expression in G93A hindlimb and diaphragm SCs with or without the 3-day NaBu treatment. RNA samples were collected from six rounds of sorting and sorted cells were seeded into three dishes as replicates. G93A HL SCs were from four male and two female mice; G93A diaphragm SCs were from two male and four female mice. ANOVA p values are also shown.

The online version of this article includes the following source data and figure supplement(s) for figure 7:

**Source data 1.** Axon guidance related genes (KEGG) identified in EOM SC signature genes cultured in growth medium.

**Source data 2.** Axon guidance related genes (KEGG) identified in EOM SC signature genes cultured in differentiation medium.

**Source data 3.** Axon guidance related genes (KEGG) identified in NaBu treatment signature genes.

**Source data 4.** qRT-PCR results for *Hmga2*, *Actn3*, *Notch3*, and *Cxcl12* in SCs of different muscle origins cultured in growth medium.

**Source data 5.** qRT-PCR results for *Hmga2*, *Actn3*, *Notch3,* and *Cxcl12* in SCs of different muscle origins cultured in differentiation medium.

**Source data 6.** qRT-PCR results for *Hmga2*, *Actn3*, *Notch3,* and *Cxcl12* in G93A diaphragm and HL SCs with or without 3-day NaBu treatment.

**Figure supplement 1.** Top functional annotations of EOM SC signature genes cultured in growth medium and differentiation medium.

hindlimb SCs cultured in differentiation medium but not in growth medium (*Figure 7C*, 2nd panel from the right). Similarly, NaBu treatment also elevated *Notch3* expression (*Figure 7D*, 2nd panel from the right) in G93A diaphragm and hindlimb SCs cultured in differential medium. Importantly, the axon guidance molecule *Cxcl12* was expressed significantly higher in EOM SCs than diaphragm and hindlimb SCs cultured in both growth and differentiation medium (*Figure 7C*, rightmost panel). NaBu treatment elevated *Cxcl12* expression as well (*Figure 7D*, rightmost panel), confirming enhanced axon attraction as a common feature shared by EOM SCs and NaBu treatment.

## AAV mediated overexpression of *Cxcl12* in myotubes promotes motor neuron axon extension

Since recombinant *Cxcl12* is known to promote axon extension (*Negro et al., 2017*), we tested whether overexpressing *Cxcl12* in myotubes via adeno-associated virus (AAV) transduction could achieve the same effect. For these experiments, we co-cultured motor neurons and SC-derived myotubes on compartmentalized microfluidic chambers (XonaChip). Because AAV vectors do not transduce satellite cells efficiently (*Arnett et al., 2014*; *Ellis et al., 2013*), we first seeded the satellite cells into one compartment of the XonaChips, then induced them to differentiate for two days before AAV transduction. Four types of AAV-transduced myotubes were used for the coculture assays: (a) WT hindlimb SC-derived myotubes transduced with AAV-CMV-eGFP; (b) G93A hindlimb SC-derived myotubes transduced with AAV-CMV-eGFP; (c) G93A hindlimb SC-derived myotubes transduced with AAV-CMV-Cxcl12-IRES-eGFP; and (d) G93A EOM SC-derived myotubes transduced with AAV-CMV-eGFP. One day after AAV transduction, rat spinal motor neurons (RSMN) were seeded into the other compartment of the XonaChips and the culture medium switched to Neurobasal Plus Medium (NBPM; *Figure 8A*). The cocultures were maintained for 9 more days. Immunostaining confirmed increased presence of Cxcl12 protein in the form of cytosolic vesicles in myotubes transduced with AAV8-CMV-Cxcl12-IRES-eGFP but not those transduced with AAV8-CMV-eGFP (*Figure 8—figure supplement 1*). These are likely vesicles for secretion (*Bebelman et al., 2020*).

By measuring the lengths of the longest RSMN neurites (presumably axons) crossing the 450 µm microgrooves, we noticed that hindlimb-SC derived myotubes from G93A mice, but not from WT littermates, appeared repulsive to distal axons. The axons stayed close to the edge of the microgrooves and even grew backward toward the neuronal compartment, which we termed 'edge effect' (*Figure 8B* and *Figure 8—figure supplement 2A*, arrows). This was rarely seen in the WT hindlimb SC-derived myotubes. In contrast, RSMN axon extension was dramatically enhanced in the co-culture system either with G93A hindlimb SC-derived myotubes overexpressing *Cxcl12*, or G93A EOM SC-derived myotubes overexpressing GFP (as control; *Figure 8B and C*). Those neurites located

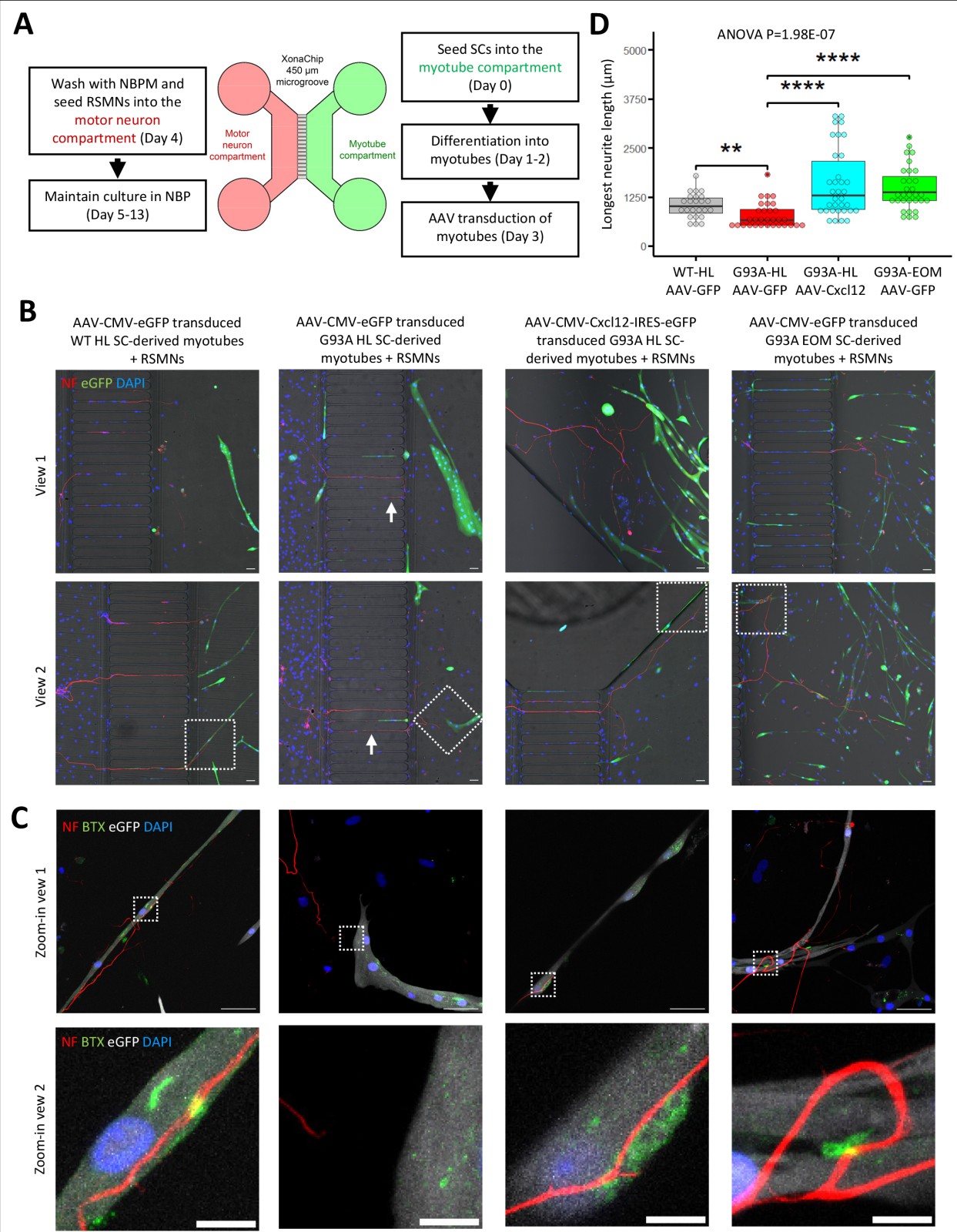

**Figure 8.** Coculture of AAV transduced SC-derived myotubes with rat spinal motor neurons in compartmentalized microfluidic chambers. (**A**) Schematic representation of XonaChip spatial configuration and coculture experiment timeline. NBPM: Neurobasal Plus Medium; RSMNs: rat spinal motor neurons. (**B**) Representative composite images of RSMNs cocultured with: 1. WT HL SC-derived myotubes transduced with AAV-CMV-eGFP; 2. G93A HL SC-derived myotubes transduced with AAV- CMV-eGFP; 3. G93A HL SC-derived myotubes transduced with AAV-CMV-Cxcl12-IRES-eGFP; 4. G93A EOM

*Figure 8 continued on next page*

*Figure 8 continued*

SC-derived myotubes transduced with AAV-CMV-eGFP. Arrows highlight RSMN neurites crossed the microgrooves but stayed close to the edges and grew backward. Also see *Figure 8—figure supplement 2A*. Boxed regions are enlarged below. Scale bars: 50 µm. (**C**) Zoom-in views of RSMN neurites innervated or missed the AChR clusters (BTX positive patches). Boxed regions are further enlarged below. Scale bars: 50 µm, 10 µm. (**D**) Measured lengths of the longest neurites derived from RSMN cells (presumably axons) crossing the 450 µm microgrooves. n=24, 31, 34,31 for the four groups, respectively (from three rounds of coculture experiments). Each dot in the box-and-dot plot represents one neurite measured. HL and EOM SCs used for co-culture experiments were all from male mice. ** p<0.01; **** p<0.0001 (t-test). ANOVA p value is also shown.

The online version of this article includes the following figure supplement(s) for figure 8:

**Figure supplement 1.** Representative views of AAV transduced G93A HL SC-derived myotubes stained with Cxcl12 and eGFP antibodies.

**Figure supplement 2.** Additional representative views of neuromuscular coculture in compartmentalized microfluidic chamber.

close to the microgrooves were often seen to closely grow along nearby myotubes (*Figure 8—figure supplement 2B*). G93A hindlimb SC-derived myotubes overexpressing *Cxcl12* occasionally were even seen to overcome the 'edge effect' and redirect the axons toward their proximities (*Figure 8—figure supplement 2B*, arrow) - a persuasive indication of chemoattractive effects of those myotubes. Meanwhile, RSMN innervation of AChR clusters (BTX positive patches) were not seen in co-cultures with G93A hindlimb SC-derived myotubes overexpressing GFP, while they were consistently observed in the other three types of AAV-transduced myotubes (*Figure 8C*).

The overall innervation incidence was too low for statistical analysis due to the sparsity of axons in the myotube compartments, thus we conducted co-culture experiments where RSMNs were added to SC-derived myotubes in a uni-compartmental co-culture setting. As shown in *Figure 9A–C* and *Figure 9—figure supplement 1A–C*, when co-cultured in a single compartment, *Cxcl12* overexpressing G93A hindlimb SC-derived myotubes exhibited increased innervation compared to GFP overexpressing controls. Furthermore, individual *Cxcl12* overexpressing EOM SC-derived myotubes were frequently innervated by multiple (3 or more) times (*Figure 9A–C* and *Figure 9—figure supplement 1D*), recapitulating the 'multi-innervation' property of EOMs in vivo (*Nijssen et al., 2017*). The extra 'anchor points' seemed to encourage better spatial alignment of the RSMN axon network to the myotubes globally in the co-cultures. This spatial alignment of RSMN axons and myotubes was not observed in co-cultures of RSMNs and hindlimb SC-derived myotubes (*without Cxcl12 overexpression*), which also did not exhibit multi-innervation (*Figure 9A–C* and *Figure 9—figure supplement 1A–C*). Under higher magnification, we often observed that axons sending branches to adjacent BTX positive patches on EOM SC-derived myotubes (*Figure 9B*, arrows). Of note, there was higher expression of several Netrin and Semaphorin family members reported to promote axon-branching in EOM SCs compared to other muscle-derived SCs cultured in differentiation medium (*Figure 7—source data 2*; *Dent et al., 2004*; *Fukunishi et al., 2011*; *Jeroen Pasterkamp et al., 2003*; *Tang and Kalil, 2005*; *Hayano et al., 2014*; *Dong et al., 2023*). This milieu, comprised of abundant axon guidance molecules, in combination with the robust self-renewability of satellite cells, may contribute to the relative EOM NMJ resistance in ALS.

## Discussion

It is known that the EOMs are complex muscles. Besides the developmental myosin isoforms, EOMs also express both adult fast and slow myosin contractile elements (*Zhou et al., 2010a*), suggesting that the sparing may not be solely linked to the fast or slow twitch nature of the muscle fibers, rather the changes in SCs may play a pivotal role in preserving the EOM function during the progression of ALS. We have demonstrated an association between the degeneration of NMJs and depletion of peri-NMJ SCs in hindlimb and diaphragm muscles in vivo in end-stage G93A mice. In contrast, EOM NMJs exhibited no signs of denervation or SCs depletion. The FACS profile of SCs isolated from EOM suggests spontaneous activation even without pathological stimulation, which is supported by previous in vivo BrdU labeling and staining results (*McLoon and Wirtschafter, 2002*). Furthermore, immunostaining, qRT-PCR and RNA-Seq analyses all imply that EOM SCs in vitro possess superior renewability better than hindlimb and diaphragm SCs, under differentiation-inducing culture conditions. Comparative functional annotation analyses of the transcriptomes highlight higher expression of axon guidance molecules in EOM SCs. Neuromuscular coculture experiments demonstrate that EOM SC-derived myotubes are more attractive to potential axons derived from rat spinal motor

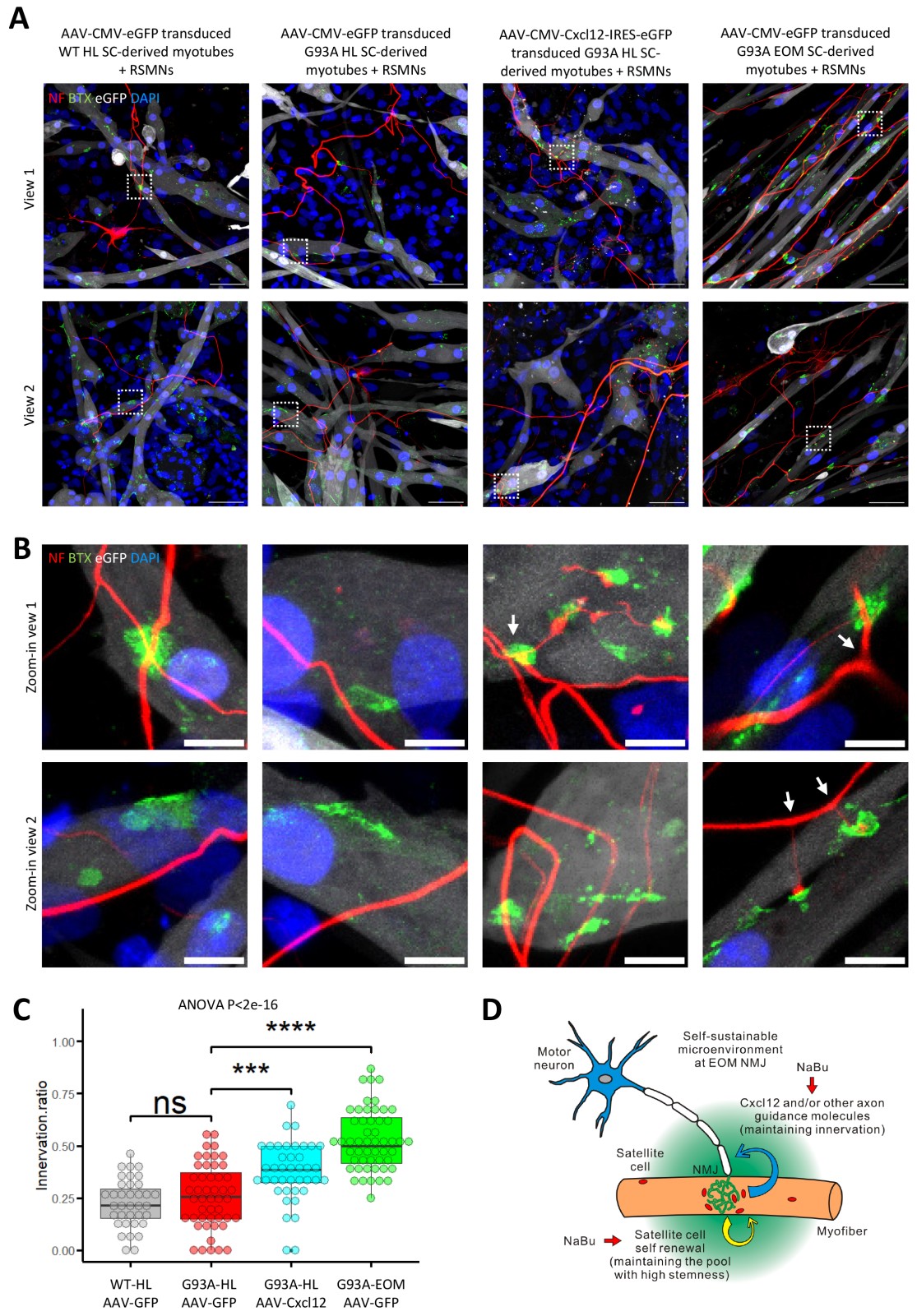

**Figure 9.** Coculture of AAV transduced SC-derived myotubes with rat spinal motor neurons within the same compartment. (**A**) Representative composite images of RSMNs cocultured with: 1. WT HL SC-derived myotubes transduced with AAV-CMV-eGFP; 2. G93A HL SC-derived myotubes transduced with AAV-CMV-eGFP; 3. G93A HL SC-derived myotubes transduced with AAV-CMV- Cxcl12-IRES-eGFP; 4. G93A EOM SC-derived myotubes transduced with AAV-CMV-eGFP. The timeline was the same as that in *Figure 8* but RSMNs were seeded on top of myotubes. Boxed regions are

*Figure 9 continued on next page*

*Figure 9 continued*

enlarged in (**B**). Scale bars: 50 µm. (**B**) Zoom-in views of innervated and non-innervated AChR clusters on myotubes having contact with RSMN neurites. Arrows highlight branching phenomena observed in RSMN neurites innervating AChR clusters. Scale bars: 10 µm. (**C**) Quantification of innervation ratios of AChR clusters (area >10 µm$^2$) in myotubes in contact with RSMN neurites. n=37, 48, 40, 52 z-stack images for the four groups, respectively (from three rounds of coculture experiments). Also see Materials and methods for detailed description. Each dot in the box-and-dot plot represents one image analyzed. HL and EOM SCs used for co-culture experiments were all from male mice. *** p<0.001; **** p<0.0001; ns, not significant (t-test). ANOVA p value is also shown. (**D**) Graphic summary of the potential mechanisms contributing to EOM NMJ integrity maintenance during ALS progression mimicked by NaBu treatment.

The online version of this article includes the following figure supplement(s) for figure 9:

**Figure supplement 1.** Additional representative views of neuromuscular coculture within the same compartment.

neurons than their hindlimb counterparts – which in fact appeared to repel the branching axons. Thus, the unique homeostasis regulation and axon attraction properties of EOM SCs may contribute to the resistance to denervation of EOM under ALS (*Figure 9D*).

Earlier report of SC quantification from muscle biopsy samples using electron microscopy detected no significant difference between ALS patients and non-ALS controls (*Ishimoto et al., 1983*). Other studies using primary cultured SCs from muscle biopsy samples implied compromised differentiation program in ALS SCs compared to controls (*Pradat et al., 2011*; *Scaramozza et al., 2014*). As to proliferation, one study reported enhanced proliferation in ALS SCs (*Scaramozza et al., 2014*), while the other cued for deficits in proliferation in ALS SCs (*Pradat et al., 2011*). The various findings may reflect variations in the disease stage of patients during sampling. For animal studies, Manzano et al measured the amount of Pax$^{7+}$ SCs in dissociated EDL and soleus myofibers from G93A mice and WT controls and observed large variations across muscle-type and disease stage (*Manzano et al., 2012*). In another study carried out by the same group, primarily cultured EDL and soleus SCs from G93A mice generally showed lower proliferation potential than the WT counterparts (*Manzano et al., 2013*).

In our whole-mount muscle imaging data and FACS profiles of isolated SCs, SC depletion, at least for the most quiescent population, is observed in hindlimb and diaphragm muscles from end-stage G93A mice. Additionally, the heterogeneity of Vcam1 levels observed in hindlimb and diaphragm SCs derived from end stage G93A mice compared to those from WT controls indicates occurrence of denervation-related pathological activation. Although this activation may transiently elevate the number of SCs, the continuous dominance of asymmetric over symmetric cell division, as well as the differentiation pressure to regenerate myofibers could gradually exhaust the quiescent stem cell pool, leading to the depletion phenotype seen in the end stage G93A mice. In contrast, the superior preservation of renewability of EOM SCs seen in differentiation medium is an effective measure against this catastrophe. However, the spontaneous activation and superior renewability of EOM SCs may not always be an advantage. As mentioned in the introduction section, EOMs are preferentially involved in some other neuromuscular disorders. In the case of mitochondrial myopathies, mitochondrial DNA defects accumulate over time, resulting in compromised oxidative phosphorylation represented by cytochrome c oxidase (COX)-deficient myofibers in patients (*Greaves et al., 2010*; *Soltys et al., 2008*). It is possible that the more frequent self-renewal and spontaneous activation of EOM SCs contribute to higher rate of mitochondrial DNA replication, leading to accelerated spreading of mitochondrial DNA defects, resulting in higher proportion of COX-deficient myofibers than other muscles, such as EDL, whose SCs remain dormant under physiological conditions.

In this study, overexpressing *Cxcl12* in G93A hindlimb SC-derived myotubes promotes motor neuron axon extension under the co-cultured condition. *Cxcl12* has been known as a critical guidance cue for motor and oculomotor axons during development (*Lieberam et al., 2005*; *Whitman et al., 2018*), as well as the migration of neural crest cells to sympathetic ganglia in the formation of sympathetic nerve system (*Kasemeier-Kulesa et al., 2010*). In adult animals, peri-synaptic Schwann cells have recently been reported to transiently produce *Cxcl12* after acute motor axon terminal degeneration (*Negro et al., 2017*). Intraperitoneal injection of antibodies neutralizing secreted Cxcl12 protein slows down the regeneration process. In contrast, the exposure of primary motor neurons to recombinant Cxcl12 protein stimulates neurite outgrowth and directional axonal extension in compartmentalized culture setup, which could be reversed by the treatment of the Cxcr4-specific inhibitor AMD3100. Published studies have demonstrated that local administration of Cxcl12 protein to neurotoxin injected hindlimb muscles of mice improved evoked junctional potentials of soleus

(*Negro et al., 2017*). Thus, the axon guidance role of Cxcl12 has been consolidated both in vitro and in vivo. Additionally, myoblasts also express *Cxcr4* and their fusion into myofibers/myotubes involves Cxcl12 guidance as well (*Griffin et al., 2010*; *Brzoska et al., 2012*; *Bae et al., 2008*). The abundant peri-NMJ SCs in EOMs expressing *Cxcl12* may achieve synergistic effects with peri-synaptic Schwann cells, improving the long-term integrity of NMJs. However, it is possible that motor neurons carrying ALS mutations may respond differently to Cxcl12-mediated axon guidance than WT motor neurons. This is a limitation of the current study, which will be investigated in future co-culture studies.

Epigenetic regulation of SC transcriptome by histone deacetylase (HDAC) inhibitors, such as butyrate (*Davie, 2003*), could modulate the myogenesis process both in vitro and in vivo (*Iezzi et al., 2002*; *Johnston et al., 1992*; *Sincennes et al., 2016*; *Moresi et al., 2015*). Butyrate is a gut-bacteria fermentation product that also acts as an energy source to improve mitochondrial respiration (*Davie, 2003*; *Donohoe et al., 2011*). Previous studies by our lab and others showed NaBu-supplemented feeding prolongs the life span of G93A mice and reverses CypD-related mitoflash phenotypes in myofibers derived from G93A mice, which implies a potential decrease of oxidative stress. Additionally, NaBu treatment also improved mitochondrial respiratory function of NSC34 motor neuron cell line overexpressing hSOD1G93A (*Li et al., 2021*; *Li et al., 2022*). NaBu treatment has been reported by other groups to directly promote neuronal survival and axonal transport by inhibiting HDAC6 (*Rivieccio et al., 2009*; *Guo et al., 2017*; *Fazal et al., 2021*; *Taes et al., 2013*). Aside from neurons, feeding NaBu also changed intestinal microbiota and intestinal epithelial permeability of G93A mice (*Zhang et al., 2017*). The effect of NaBu treatment is certainly multi-faceted. Yet previously it was not known whether the beneficiary effects of NaBu are linked to any phenotypic changes of SCs, and whether NaBu treatment induced epigenetic modulations similar to that of EOM SCs. The study here revealed its beneficiary role on the preservation of NMJ integrity and peri-NMJ SCs affected in hindlimb and diaphragm muscles by ALS progression. NaBu treatment of G93A hindlimb and diaphragm SCs in culture induced transcriptomic changes mimicking some features seen in EOM SCs, including the upregulation of activation and quiescence signature genes and the downregulation of differentiation signature genes, indicating NaBu improved the renewability of G93A hindlimb and diaphragm SCs. Additionally, some axon guidance molecules, particularly *Cxcl12*, were also induced by NaBu application (*Figure 9D*). Thus, NaBu-induced EOM SC-like transcriptomic pattern could be a crucial contributor to its therapeutic effect against ALS progression. Combining NaBu treatment with application on top of *Cxcl12* overexpression may have additive therapeutic benefits to slow ALS progression.

Overall, our studies have defined several mechanisms which (1) may explain why EOMs are relatively spared in ALS, and (2) provides a rationale for targeting these differences as a therapeutic strategy. Furthermore, analysis of these signatures in patient muscle biopsies could be used to select subsets of ALS patients who may be most likely to benefit from therapies targeting this signaling network, and to provide possible 'response biomarkers' in pre-clinical and clinical studies. As a cocktail of sodium phenylbutyrate (a derivative of NaBu) and Taurursodiol (an antioxidant) under the Commercial name of Relyvrio has been recently approved in US and Canada as a new ALS therapy (*Paganoni et al., 2020*), our study could help gain more insights into the molecular mechanisms related to this type of ALS therapy.

## Materials and methods

**Key resources table**

| Reagent type (species) or resource | Designation | Source or reference | Identifiers | Additional information |
|---|---|---|---|---|
| Strain, strain background (*Mus musculus*) | B6SJLF1/J | Jackson Laboratory | Stock # 100012 | WT mice<br>Both male and female<br>Age up to 4 months |
| Strain, strain background (*Mus musculus*) | B6SJL-Tg (SOD1*G93A) | Jackson Laboratory | Stock # 002726 | G93A mice<br>Both male and female<br>Age up to 4 months |

*Continued on next page*

*Continued*

| Reagent type (species) or resource | Designation | Source or reference | Identifiers | Additional information |
|---|---|---|---|---|
| Cell line (*Mus musculus*) | Primary cultured mouse satellite cells | This paper | | FACS-isolated from skeletal muscles dissected from WT or G93A mice, including both male and female. |
| Cell line (*Rattus norvegicus*) | Primary cultured rat spinal motor neurons (RSCMNs) | iXCells | SKU # 10RA-033 | Isolated from D16 rat embryo spinal cord; Negative for mycoplasma, bacteria, yeast, and fungi. |
| Chemical compound, drug | Sodium butyrate | SAFC | Product # ARK2161 | 2% in drinking water for animal feeding; 0.5 mM in culture medium for cell treatment. |
| Transfected construct | AAV8-CMV-eGFP | Vector Biolabs | Item Code 7777 | Adeno-associated virus to transfect and express eGFP. |
| Transfected construct (*Mus musculus*) | AAV8-CMV-mCXCL12-IRES-eGFP | Vector Biolabs | Item Code 7000 | Adeno-associated virus to transfect and express mouse CXCL12 and eGFP. |
| Peptide, recombinant protein | Collagenase II | Worthington | Cat # LS004176 | Final concentration 0.26% |
| Peptide, recombinant protein | Dispase II | Sigma Aldrich | Cat # D46931G | Final concentration 0.24% |
| Peptide, recombinant protein | Hyaluronidase | Worthington | Cat # LS002592 | Final concentration 0.16% |
| Peptide, recombinant protein | DNase I | Worthington | Cat # LS002139 | Final concentration 0.04% |
| Antibody | Anti-mouse Vcam1-PE (Rat monoclonal IgG2a) | BioLegend | Clone 429 (MVCAM.A) Cat # 105713 | 600 ng/$10^6$ cells for FACS isolation |
| Antibody | Anti-mouse CD31-APC (Rat monoclonal IgG2a) | BioLegend | Clone 390 Cat #102409 | 400 ng/$10^6$ cells for FACS isolation |
| Antibody | Anti-mouse CD45-APC (Rat monoclonal IgG2b) | BioLegend | Clone 30-F11 Cat # 103111 | 400 ng/$10^6$ cells for FACS isolation |
| Antibody | Anti-mouse Sca-1-APC (Rat monoclonal IgG2a) | BioLegend | Clone D7 Cat # 108111 | 200 ng/$10^6$ cells for FACS isolation |
| Antibody | Anti-chicken Pax7 (Mouse monoclonal IgG1) | Santa Cruz Biotechnology | Cat # sc-81648 | IF (1:100) |
| Antibody | Anti-mouse MyoD (Mouse monoclonal IgG2b) | Santa Cruz Biotechnology | Cat # sc-377460 | IF (1:100) |
| Antibody | Anti-mouse Ki-67 (Rabbit monoclonal IgG) | Cell Signaling Technology | Cat # 12202 S | IF (1:300) |
| Antibody | Anti-chicken MHC (Mouse monoclonal IgG2b) | Developmental Studies Hybridoma Bank | Cat # MF20 | IF (1:200) |
| Antibody | Anti-rat NF-M (Mouse monoclonal IgG1) | Developmental Studies Hybridoma Bank | Cat # 2H3 | IF (1:300) |
| Antibody | Anti-GFP (Mouse monoclonal IgG2a) | Novus | Clone 4B10B2 Cat # NBP2-22111AF488 | IF (1:250) |
| Antibody | Anti-rat Synaptophysin (Rabbit polyclonal IgG) | Thermo Fisher Scientific | PA11043 | IF (1:400) |
| Antibody | Anti-human Cxcl12 (Mouse monoclonal IgG1) | R&D systems | Clone 79018 Cat # MAB350 | IF (1:200) |
| Other | AF 488 conjugated α-Bungarotoxin | Thermo Fisher Scientific | Cat # B13422 | IF (1:1000) |

*Continued on next page*

*Continued*

| Reagent type (species) or resource | Designation | Source or reference | Identifiers | Additional information |
|---|---|---|---|---|
| Other | DAPI in water | Biotium | Cat # 40043 | IF (1:8000) |
| Other | Methyl green 0.1% aqueous solution | Alfa Aesar | Cat # 42747 | Diluted 1:500 to stain dead cells for FACS analysis |
| Sequence-based reagent | qPCR primers for mouse *Actn3* | Sigma Aldrich | | Fwd: AACAGCAGCGGAAAACCTTCA Rev: GGCTTTATTGACATTGGCGATTT |
| Sequence-based reagent | qPCR primers for mouse *Cxcl12* | Sigma Aldrich | | Fwd: TGCATCAGTGACGGTAAACCA Rev: TTCTTCAGCCGTGCAACAATC |
| Sequence-based reagent | qPCR primers for mouse *Gapdh* | Sigma Aldrich | | Fwd: AGGTCGGTGTGAACGGATTTG Rev: TGTAGACCATGTAGTTGAGGTCA |
| Sequence-based reagent | qPCR primers for mouse *Hmga2* | Sigma Aldrich | | Fwd: GAGCCCTCTCCTAAGAGACCC Rev: TTGGCCGTTTTTCTCCAATGG |
| Sequence-based reagent | qPCR primers for mouse *Notch3* | Sigma Aldrich | | Fwd: GGTAGTCACTGTGAACACGAGG Rev: CAACTGTCACCAGCATAGCCAG |
| Sequence-based reagent | qPCR primers for mouse *Scn5a* | Sigma Aldrich | | Fwd: ATGGCAAACTTCCTGTTACCTC Rev: CCACGGGCTTGTTTTTCAGC |

## Animals

All animal experiments were carried out in accordance with the recommendations in the *Guide for the Care and Use of Laboratory Animals* of the National Institutes of Health. The protocol on the usage of mice was approved by the Institutional Animal Care and Use Committee of the University of Texas at Arlington (approved IACUC protocol #A19.001). WT mice used in this study were of B6SJL background. The ALS transgenic mouse model (hSOD1G93A) with genetic background of B6SJL was originally generated by Drs. Deng and Siddique's group at Northwestern University and deposited to the Jackson Lab as B6SJL-Tg (SOD1*G93A) (*Gurney et al., 1994*). For the butyrate feeding experiment, the animals were fed with 2% NaBu (SAFC ARK2161) from the age of 3 months (at the ALS disease onset) and lasting for 4 weeks.

## Primary culture of satellite cells isolated by fluorescence-activated cell sorting

The hindlimb muscles including tibialis anterior (TA), extensor digitorum longus (EDL), gastrocnemius, and quadriceps were dissected from one mouse for each round of sorting. For diaphragm, both the crural and costal domains were dissected from one mouse for each round of sorting. For EOMs, the four rectus muscles in the eye sockets were dissected from 5 to 6 mice of the same gender for each round of sorting (*Eckhardt et al., 2019*). The collected muscles were placed in 0 mM $Ca^{2+}$ Ringer's solution. Excessive connective tissue, tendons, fat, blood clots and vessels were removed as much as possible. After cleanup, the muscles were transferred into the digestion media (DMEM containing 5% FBS, 1% Pen/Strep and 10 mM HEPES) and minced into small blocks using scissors. Collagenase II (Worthington LS004176,≥125 units/mg), dispase II (Sigma D46931G,≥0.5 units/mg), hyaluronidase (Worthington LS002592,≥300 USP/NF units/mg) and Dnase I (Worthington LS002139,≥2,000 Kunitz units/mg) were added at the final concentration of 0.26%, 0.24%, 0.16%, and 0.04% (weight/volume), respectively. The digestion system was placed in an orbital shaker running at 70 rpm at 37 °C for 45 min. The digested muscles were triturated 10 times with a 20-gauge needle attached to a 5 ml syringe. Afterwards the triturated mixture was pipetted onto a pre-wetted 100 μm strainer and centrifuge at 100 × *g* for 2.5 min to precipitate the bulky myofiber debris. The supernatant was transferred onto a pre-wetted 40 μm strainer and cells were collected by centrifuged at 1200 × *g* for 6.5 min. The cells were resuspended in sorting media (PBS containing 0.3% BSA, 10 mM HEPES and 1 mM EGTA) and stained with antibodies or dyes at 4 °C for 45 min with shaking. The antibodies used include PE anti-mVcam1 (Biolegend 105713, 600 ng/10⁶ cells), APC anti-mCD31 (Biolegend 102409,

400 ng/10$^6$ cells), APC anti-mCD45 (Biolegend 103111, 400 ng/10$^6$ cells) APC anti-mSca-1 (Biolegend 108111, 200 ng/10$^6$ cells). This antibody combination followed a previous published protocol (*Liu et al., 2015a*). Vcam1 is a SC marker, CD31 is a endothelial cell marker, CD45 is a hematopoietic cell marker and Sca-1 is a mesenchymal stem cell marker (*Liu et al., 2015a*). For sorting, our target population is VCAM$^+$CD31$^-$CD45$^-$Sca-1$^-$. When needed, the cells were stained with methyl green (2 μg/ml) to check mortality rate. As a positive control, half of the cells were transferred to a new Eppendorf tube and methanol (precooled at –20 °C) was added dropwise with gentle mixing and left on ice for 5 min, the dead cells were wash twice before being mixed with the rest half of cells for methyl green staining. The sorting process was conducted using BD FACS Melody Cell Sorter using a 100 μm nozzle. The excitation laser and emission filter setup for methyl green is the same as APC. The flow rate was tuned between 1–7 to adjust the event rate to 2000–4000/sec. Gating conditions of FSC-A/SSC-A, SSC-H/SSC-W, FSC-H/FSC-W, Vcam1/CD31_CD45_Sca-1 and Vcam1/Methyl green plots were shown in *Figure 3—figure supplement 1* and *Figure 3—figure supplement 2*. The sorted cells were collected in growth media (DMEM containing 20%FBS, 1% Pen/Strep, 1% chick embryonic extract and 10 mM HEPES). Afterwards the cells were seeded into laminin (Corning 354232) -coated plates or glass-bottom dishes containing growth media (without HEPES). Culture media was changed on the third day and the fourth day. To induce differentiation, cells were washed three times with PBS and cultured in differentiation media (DMEM containing 2% horse serum) for 2 more days.

## Neuromuscular coculture assay

For neuromuscular coculture in compartmentalized chambers (XonaCHIPS XC450), XoanCHIPS were coated with poly-L-ornithine (0.5 mg/ml) for 1 hr at 37 °C followed by laminin (0.1 mg/ml) for 1 hr at 37 °C. Primary cultured satellite cells were seeded at 1x10$^5$ into the myotube compartment in growth medium for attachment overnight and then switched to differentiation medium and cultured for 2 more days. Afterwards AAV8-CMV-eGFP (Vector Biolabs, 1x10$^{11}$ GC/ml) or AAV8-CMV-Cxcl12-IRES-eGFP (Vector Biolabs, 1.5x10$^{11}$ GC/ml, the *Cxcl12* gene is of mouse origin) transduction was conducted for one day. The usage of higher titer for the second virus is for matching GFP signal intensity, as IRES-dependent expression is significantly lower than those directly driven by CMV (*Mizuguchi et al., 2000*). The chambers were washed with Neurobasal Plus Medium twice before seeding rat spinal motor neurons into the neuronal compartment (1x10$^5$). The coculture was kept in Neurobasal Plus Medium for 9 more days and half medium change was conducted every 2 days.

For neuromuscular coculture in the same glass-bottom chamber, ight-well chambered cover glasses (Cellvis C8-1.5H-N) were coated with poly-L-ornithine and laminin as described above. Primary cultured satellite cells were seeded at 1x10$^5$/chamber in growth medium for attachment overnight and then switched to differentiation medium and cultured for 2 more days. AAV transduction was conducted for one day as described above. The infected myotubes were washed with Neurobasal Plus Medium (Thermo Fisher A3582901, with B-27 and GlutaMax supplement) before the addition of rat spinal motor neurons (4x10$^3$/chamber, iXCells 10RA-033) to the same chamber. The coculture was kept in Neurobasal Plus Medium for 9 more days and half medium change was conducted every 2 days.

## Immunofluorescence (IF) and imaging of whole mount muscle samples

EDL, soleus and EOM (the four rectus muscles) were dissected out and fixed in 4% paraformaldehyde (PFA) for 1 hr. Afterwards the samples were washed with PBS containing 1% glycine once and two more times with PBS. Further fixation and permeabilization was done with methanol at –20 °C for 10 min. For diaphragm, the costal muscles and their surrounding rib cages were dissected and immersed in methanol directly for fixation (PFA fixation was skipped because it made the diaphragm harder to flatten and introduced more non-specific noises in staining). The samples were rehydrated and washed with PBS. The epimysium and tendons were removed as much as possible. EDL and soleus were further split into smaller bundles along the longitudinal axis. Then the samples were immersed in blocking buffer (PBS containing 2% BSA, 2% horse serum, 0.1% Tween-20, 0.1% Triton X-100 and 0.05% sodium azide) overnight at 4 °C. Next day the samples were incubated with primary antibodies for 1–2 days. After washing with PBS for three times, the samples were incubated with Alexa Fluor labeled secondary antibodies (Thermo Fisher 1:1000) and Alexa Fluor 488 conjugated α-Bungarotoxin (Thermo Fisher B13422, 1:1000) for 1–2 days. The samples were then washed with

PBS, counterstained with DAPI and mounted in antifade mounting media (20 mM Tris, 0.5% N-propyl gallate, 80% glycerol) for imaging. Primary antibodies used: neurofilament (DSHB 2H3 concentrate, 1:250), synaptophysin (Thermo Fisher PA11043, 1:400), Pax7 (SCBT sc-81648, 1:100), Cxcl12 (R&D MAB350, 1:200).

Z-stack scans of glycerol-cleared whole muscles were obtained using 40 x high working distance lens of Leica TCS SP8 confocal microscope. The z- stacks were compacted into 2D images by maximal intensity projection. To quantify SC number surrounding NMJs, circles (75 µm in diameter) were drawn around each NMJ in Image J. The DAPI channel was used as a mask to filter out nonspecific staining of Pax7 in the extracellular matrix.

## Immunofluorescence and imaging of cultured myoblasts/myotubes and motor neurons

The cells were fixed in 4% PFA for 10 min and then washed with 1% glycine and PBS. Further fixation and permeabilization was done either with methanol at –20 °C for 10 min or 0.25% TritonX-100 for 15 min. The samples were then washed with PBS and immersed in blocking buffer for 40 min. Primary antibody incubation was done at 4 °C overnight. Next day, the cells were washed with PBS and incubated with Alexa Fluor labeled secondary antibodies (1:1000) for 2 hr at room temperature, counterstained with DAPI and mounted in antifade mounting media for imaging. Primary antibodies used: Alexa Fluor 488 conjugated α-Bungarotoxin (Thermo Fisher B13422, 1:1000), Pax7 (SCBT sc-81648, 1:100), MyoD (SCBT sc-377460, 1:100), Ki-67 (CST 12,202 S, 1:300), MHC (DSHB MF20 concentrate, 1: 200), neurofilament (DSHB 2H3 concentrate, 1:250), eGFP (Novus NBP-22111AF488, 1:250).

For quantifying innervation ratio, AChR clusters (BTX patches) larger than 10 µm$^2$ were counted on myotubes that have contact with RSMN neurites. Those myotubes having 0 pixel overlap with RSMN neurites were not included in measurement. The ratio of AChR clusters having overlaps with the RSMN neurites after visual inspection of the whole z-stacks of each view were calculated. The innervation ratio of each view is one dot in the box-and-dot plot in *Figure 9C*. The measurements were done for 3 rounds of coculture assays.

## RNA extraction and qRT-PCR

For cultured SCs, cell culture medium was removed before RNA extraction and Trizol reagent was added to the well before scratching cells off using pipette tips. The homogenate was transferred into tubes containing 5 PRIME Phase Lock Gel (Quantabio 2302830), and 1/10 volume of 1-bromo-3-chloroporpane (Alfa Aesar A11395) was added. The samples were shaken vigorously for 12 s and centrifuged for 15 min at 16,000 × g. Afterwards, the upper phase was transferred to another Eppendorf tube and isopropanol (1/2 volume of the initial homogenate) was added for precipitation overnight at −20 °C. Next day, after 15 min centrifugation at 16,000 g the precipitate was washed briefly with 75% ethanol. After another 5 min centrifugation, the ethanol was removed. The precipitate was briefly dried and resuspended in RNase-free water. For whole muscle samples, the freshly dissected muscles were snap-frozen in liquid nitrogen, kept at −80 °C freezer overnight and then soaked in RNALater ICE (Thermo Fisher AM7030) for 2–3 days at −20 °C. Before RNA extraction, leftover tendons and neurites were removed and muscles were transferred to 1 ml Trizol. Homogenization was conducted in FastPrep-24 Classic bead beating grinder (MP Biomedicals 116004500). The tissue homogenate was transfer to centrifuge tubes containing phase lock gel (QuantaBio 2302830) and 1/10 volume of 1-bromo-3-chloropropane was added. The tubes were hand shaken for 12 s, left at bench top for 2 min and then centrifuged at 16,000 × g at 4 °C for 15 min. The upper phase was transferred to a new centrifuge tube and mixed with equal volume of ethanol. The following steps of RNA purification was conducted with Direct-zol RNA Miniprep Plus kit (Zymo Research R2070). RNA concentration was measured with Quantus Fluorometer (Promega E6150). RNA quality was assessed by RNA ScreenTape analysis at Genomics Core Facility at the University of North Texas Health Science Center. Only RNAs with RIN ≥9 and 28 s/18s≥1 were used for qRT-PCR or RNA-Seq. Reverse transcription was done using Promega GoScript Reverse Transcription system. The qPCR reactions used Maxima SYBR Green/ROX qPCR Master Mix (Thermo Fisher) and were carried out by the StepOnePlus Real-Time PCR system (Thermo Fisher). Relative quantification (RQ) of gene expression was generated by ΔΔCt method. The sequences of primers used for qPCR were listed in Key Resource Table.

## Bulk RNA-Seq and analysis

Non-stranded mRNA sequencing was carried out by BGI Americas Corporation. The average number of clean reads per sample was 45.16 million (ranging from 44.11 to 48.51 million). The average percentage of clean reads was 93.32% (ranging from 92.42% to 93.97%). The average percentage of aligned reads (to Genome Reference Consortium Mouse Build 38 patch release 6) was 96.01% (ranging from 95.40% to 96.67%) and the average percentage of uniquely aligned reads was 86.46% (ranging from 85.12% to 87.63%). To identify differentially expressed genes (DEGs) between different samples (using Poisson distribution method to compare uniquely mapped reads of the gene), only genes with $|log2FC|{\geq}0.4$ and FDR $\leq$0.001 were considered. To identify DEGs between different groups (containing more than one sample, using DESeq2 method), only genes with $|log2FC|{\geq}0.4$ and Qvalue $\leq$0.05 were considered. PCA analysis, hierarchical clustering, functional annotation analysis (KEGG pathway, GO_C, GO_P) were performed in Dr. TOM platform of BGI. Dot plots showing the distribution of activation, differentiation and quiescence signature genes were generated in ggplot2 in R. Fastq files and the excel file containing TPM values has been uploaded to GEO database with record number GSE249484.

## Data analysis and statistics

The experimenters were not blinded to the samples in data collection and analysis. Box-and-dot plots were generated using the open-source ggplot2 data visualization package for the statistical programming language R. The lower hinge, median line and upper hinge correspond to the first, second and third quartiles. The lower and upper whiskers extend from the hinges to the largest value no further than 1.5 times of inter-quartile range (distance between the first and third quartiles). Data beyond the end of the whiskers are outlying points. Two independent sample t-tests were carried out by adding the stat_compare_means function. For multi-group data, one-way ANOVA -values were generated using the R function aov(). For each two-group comparison, we used two independent sample t-tests and used brackets to highlight the groups compared.

## Acknowledgements

We thank Dr. Kimberly Bowles at Life Science Core Facility of UT Arlington for the assistance in cell sorting experiments. This study has been supported by grants from NIH (R01NS105621 to JZ, R01NS129219 to JZ, JM, LWO, and R01AG071676 To JM and JZ) and the Department of Defense AL170061(W81XWH1810684) to JZ.

## Additional information

### Funding

| Funder | Grant reference number | Author |
| --- | --- | --- |
| National Institutes of Health | R01NS105621 | Jingsong Zhou |
| National Institutes of Health | R01NS129219 | Lyle W Ostrow Jianjie Ma Jingsong Zhou |
| National Institutes of Health | R01AG071676 | Jianjie Ma Jingsong Zhou |
| U.S. Department of Defense | AL170061(W81XWH1810684) | Jingsong Zhou |

The funders had no role in study design, data collection and interpretation, or the decision to submit the work for publication.

### Author contributions

Ang Li, Conceptualization, Data curation, Formal analysis, Validation, Investigation, Visualization, Methodology, Writing – original draft, Project administration, Writing – review and editing; Jianxun

Yi, Conceptualization, Data curation, Methodology, Writing – review and editing; Xuejun Li, Li Dong, Data curation, Validation, Investigation, Methodology; Lyle W Ostrow, Jianjie Ma, Conceptualization, Funding acquisition, Writing – review and editing; Jingsong Zhou, Conceptualization, Resources, Supervision, Funding acquisition, Validation, Investigation, Visualization, Methodology, Writing – original draft, Project administration, Writing – review and editing

## Author ORCIDs

Ang Li ⓘ http://orcid.org/0000-0002-8784-4702
Lyle W Ostrow ⓘ https://orcid.org/0000-0001-7158-5243
Jingsong Zhou ⓘ http://orcid.org/0000-0002-5684-7177

## Ethics

All animal experiments were carried out in accordance with the recommendations in the Guide for the Care and Use of Laboratory Animals of the National Institutes of Health. The protocol on the usage of mice was approved by the Institutional Animal Care and Use Committee of the University of Texas at Arlington (approved IACUC protocol # A19.001).

Joint Public Review: https://doi.org/10.7554/eLife.92644.4.sa1
Author response https://doi.org/10.7554/eLife.92644.4.sa2

---

# Additional files

## Supplementary files

• MDAR checklist

## Data availability

Fastq files and the excel file containing TPM values are available at GEO database with record number GSE249484. All data generated or analysed during this study are included in the manuscript and supporting files; source data files have been provided for Figures 1, 2, 3, 6, 7. Figure 1-source data 1 and 2, Figure 2-source data 1, Figure 3-source data 1, Figure 6-source data 1-6, Figure 7-source data 1-6, contain the numerical data used to generate the figures.

The following dataset was generated:

| Author(s) | Year | Dataset title | Dataset URL | Database and Identifier |
|---|---|---|---|---|
| Li A, Yi J, Li X, Dong L, Ostrow LW, Zhou J | 2023 | Distinct transcriptomic profile of satellite cells contributes to preservation of neuromuscular junctions in extraocular muscles of ALS mice | https://www.ncbi.nlm.nih.gov/geo/query/acc.cgi?acc=GSE249484 | NCBI Gene Expression Omnibus, GSE249484 |

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
