## [Editor Report · eLife assessment]

The manuscript by Jingsong Zhou and colleagues uncovers why the extraocular muscles (EOMs) are preserved while other muscles undergo degenerative changes in amyotrophic lateral sclerosis (ALS). In this work, the authors have used a mouse model of familial ALS that carries a G93A mutation in the Sod1 gene to demonstrate that NaBu treatment partially restores the integrity of NMJ in the limb and diaphragm muscles of G93A mice. The findings of the study offer **important** information that EOMs are spared in ALS because they produce protective factors for the NMJ and, more specifically, factors secreted by EOM-derived satellite cells. While most of the experimental approaches are **convincing**, the use of sodium butyrate (NaBu) in this study needs further investigation, as NaBu might have a variety of biological effects. Overall, this work may help develop future therapeutic interventions for patients with ALS.

---

## [Referee Report · Joint Public Review]

Summary:

In their paper Li et al. investigate the transcriptome of satellite cells obtained from different muscle types including hindlimb, diaphragm and extraocular muscles (EOM) from wild type and G93A transgenic mice (end stage ALS) in order to identify potential factors involved in the maintenance of the neuromuscular junction. The underlying hypothesis being that since EOMs are largely spared from this debilitating disease, they may secrete NMJ-protective factors. The results of their transcriptome analysis identified several axon guidance molecules including the chemokine Cxcl12, which are particularly enriched in EOM-derived satellite cells. Transduction of hindlimb-derived satellite cells with AAV encoding Cxcl12 reverted hindlimb-derived myotubes from the G93A mice into myotubes sharing phenotypic characteristics similar to those of EOM-derived satellite cells. Additionally, the authors were able to demonstrate that EOM-derived satellite cell myotube cultures are capable of enhancing axon extensions and innervation in co-culture experiments.

Strengths:

The strength of the paper is that the authors successfully isolated and purified different populations of satellite cells, compared their transcriptomes, identified specific factors release by EOM-derived satellite cells, overexpressed one of these factors (the chemokine Cxcl12) by AAV-mediated transduction of hindlimb-derived satellite cells. The transduced cells were then able to support axon guidance and NMJ integrity. They also show that administration of Na butyrate to mice decreased NMJ denervation and satellite cell-depletion of hind limbs. Furthermore, addition of Na Butyrate to hindlimb derived satellite cell myotube cultures increased Cxcl12 expression. These are impressive results providing important insights for the development of therapeutic targets to slow the loss on neuromuscular function characterizing ALS.

Comments on latest version:

The authors have sufficiently acknowledged and discussed the limitations of experiments involving NaBu treatment. The authors have also addressed the use of AAV-mediated delivery of Cxcl12.

---

## [Author Response]

The following is the authors’ response to the previous reviews.

**eLife assessment**
The manuscript by Jingsong Zhou and colleagues tries to uncover the reasons for the resistance of extraocular muscles (EOMs) to degenerative changes induced by amyotrophic lateral sclerosis (ALS). The findings of the study offer valuable information that EOMs are spared in ALS because they produce protective factors for the NMJ and, more specifically, factors secreted by EOM-derived satellite cells. While most of the experimental approaches are convincing, the use of sodium butyrate (NaBu) in this study needs further investigation, as NaBu might have a variety of biological effects. Overall, this work may help develop future therapeutic interventions for patients with ALS.

We agree with the editor that NaBu have a variety of biological effects that require further investigation. Our team previously have explored the effect of NaBu treatment on intestinal microbiota and intestinal epithelial permeability (DOI: 10.1016/j.clinthera.2016.12.014), on the mitochondrial respiratory function of NSC-34 motor neuron cell line overexpressing hSOD1G93A (DOI: 10.3390/biom12020333) and on the mitochondrial function of skeletal muscle myofibers of G93A mice (DOI: 10.3390/ijms22147412). Other research teams have also explored the role of NaBu (or HDAC inhibition) in neuronal survival and axonal transport (DOIs: 10.1073/pnas.0907935106; 10.1038/s41467-017-00911-y; 10.15252/embj.2020106177; 10.1093/hmg/ddt028).

Since the theme of this manuscript is the transcriptomic characteristics of EOM SCs, to include data of how NaBu affect cellular/molecular processes of other tissues will somewhat deviate from the theme. It would be more appropriate to develop a separate manuscript focusing on other tissues.

We appreciate the feedback from the Editors and reviewers. We realized that our previous description on butyrate’s beneficial role might be overstated in the Abstract Section. We have made two changes to avoid potential overstatement of our finding: (1) We modified the Abstract to state that “the NaBu-induced transcriptomic changes resembling the patterns of EOM SCs “may contribute to” (instead of “underlie”) the beneficial effects observed in G93A mice” (Page 1, Line 29); (2) We have edited the corresponding paragraph in the Discussion section to emphasize that the effect of NaBu treatment is multi-faceted (Page 11, Line 459-461).

**Recommendations for the authors:**

**Reviewer #3 (Recommendations For The Authors):**
line 388-389. The sentence has been corrected but is still not clear. What do the authors mean by ".....resulting in higher proportion of COX-deficient myofibers than other muscles». What other muscles do they refer to?

Other muscles refer to muscles whose stem cells remain dormant under physiological conditions (uninjured, innervated), such as EDL. We have edited the sentence accordingly. (Page 10, Line 431-432)

In reference to the results shown in Fig. 2, 7, 8 and 9. Since the experimenters were not blinded, this should be explicitly stated in the Methods section.

We have added the disclaimer in the current “Data analysis and statistics” section in Methods as follows: “The experimenters were not blinded to the samples in data collection and analysis.” (Page 15, Line 636)

Figure 7 C has been amended but now the inserted ANOVA values interfere with the correct visualization of Fig. 7D, can panels D be moved down so that they are better separated from panels in Fig. 7C

Thanks for the comment and we have edited Figure 7 accordingly.

**Reviewer #4 (Recommendations For The Authors):**
The authors have revised the manuscript per the reviewer's comments in this study. While most of the concerns were addressed, a few concerns remain.The molecular basis of how AAV-mediated delivery of Cxcl12 improves the phenotype of satellite cells is still unclear.

Thanks for the comment. As one of the earliest discovered chemokines, the chemotactic role of Cxcl12-Cxcr4 axis on cells and cellular processes (such as axons) has been comprehensively investigated by different functional assays from overexpression to protein application to inhibitor application to knockdown by shRNAs in different types of tissues. To list a few examples, the establishment of the correct routing trajectories of mammalian motor axons and oculomotor axons during embryonic development (DOIs: 10.1016/j.neuron.2005.08.011; 10.1167/iovs.18-25190). The regeneration of injured motor axon terminals guided by terminal Schwann cells in adult mice (DOI: 10.15252/emmm.201607257). The migration of neural crest cells to sympathetic ganglia in the formation of sympathetic nerve system during embryogenesis (DOI: 10.1523/JNEUROSCI.0892-10.2010). The migration of myoblasts in the process of fusion into myotubes (DOIs: 10.1242/jcs.066241; 10.1111/boc.201200022; 10.1074/jbc.M706730200).

Because the existence of so many detailed mechanistic studies, our goal for this manuscript is not to identify a novel mechanism of how Cxcl12-mediated chemotaxis is achieved. Rather, we used it as one of the proof-of-concept mechanisms contributing to the resistance of EOMs against ALS and benefits of NaBu treatment. Certainly, it is not the sole mechanism.

To address the reviewer’s concern, we have expanded discussion about the previous studies regarding the chemotactic effect of Cxcl12 in the discussion section. (Page 10, Line 435-436, Page 11, Line 445-446)

The NaBu experiments may need additional support from other approaches. NaBu effects may not be directly related to satellite cells or muscle cells. Thus, the animal experiment results need to be carefully interpreted.

We agree that NaBu have a variety of biological effects that require further investigation. Our team previously have explored the effect of NaBu treatment on intestinal microbiota and intestinal epithelial permeability (DOI: 10.1016/j.clinthera.2016.12.014), on the mitochondrial respiratory function of NSC-34 motor neuron cell line overexpressing hSOD1G93A (DOI: 10.3390/biom12020333) and on the mitochondrial function of skeletal muscle myofibers of G93A mice (DOI: 10.3390/ijms22147412). Other research teams have also explored the role of NaBu (or HDAC inhibition) in neuronal survival and axonal transport (DOIs: 10.1073/pnas.0907935106; 10.1038/s41467-017-00911-y; 10.15252/embj.2020106177; 10.1093/hmg/ddt028).

Since the theme of this manuscript is the transcriptomic characteristics of EOM SCs, to include data of how NaBu affect cellular/molecular processes of other tissues will somewhat deviate from the theme. It would be more appropriate to develop a separate manuscript specifically addressing the impact of NaBu on other tissues.

We appreciate the feedback from the reviewers. We realized that our previous description on butyrate’s beneficial role might be overstated in the Abstract Section. In response, we have made two changes to avoid potential overstatement of our finding: (1) We modified the Abstract to state that “the NaBu-induced transcriptomic changes resembling the patterns of EOM SCs “may contribute to” (instead of “underlie”) the beneficial effects observed in G93A mice” (Page 1, Line 29); (2) We edited the corresponding paragraph in the Discussion section to emphasize that the effect of NaBu treatment is multi-faceted (Page 11, Line 459-461).